# The Importance of Antecedent Vegetation and Drought Conditions as Global Drivers of Burnt Area

Alexander Kuhn-Régnier[1,2], Apostolos Voulgarakis[1,2,3], Peer Nowack[2,4,5], Matthias Forkel[6],
I. Colin Prentice[1,7], and Sandy P. Harrison[1,8]

[1]Leverhulme Centre for Wildfires, Environment, and Society
[2]Department of Physics, Imperial College London, UK
[3]School of Environmental Engineering, Technical University of Crete, Chania, Greece
[4]Grantham Institute and the Data Science Institute, Imperial College London, UK
[5]Climatic Research Unit, School of Environmental Sciences, University of East Anglia, UK
[6]Environmental Remote Sensing Group, TU Dresden, Dresden, Germany
[7]Department of Life Sciences, Imperial College London, UK
[8]Geography and Environmental Science, University of Reading, UK

**Correspondence:** Alexander Kuhn-Régnier (alexander.kuhn-regnier14@imperial.ac.uk)

**Abstract.** The seasonal and longer-term dynamics of fuel accumulation affect fire seasonality and the occurrence of extreme wildfires. Failure to account for their influence may help to explain why state-of-the-art fire models do not simulate the length and timing of the fire season or interannual variability in burnt area well. We investigated the impact of accounting for different timescales of fuel production and accumulation on burnt area using a suite of random forest regression models that included the immediate impact of climate, vegetation, and human influences in a given month, and tested the impact of various combinations of antecedent conditions in four productivity-related vegetation indices and in antecedent moisture conditions. Analyses were conducted for the period from 2010 to 2015 inclusive. Inclusion of antecedent vegetation conditions representing fuel build-up led to an improvement of the global, climatological out-of-sample $R^2$ from $0.579$ to $0.701$, but the inclusion of antecedent vegetation conditions on timescales $\geq 1$ yr had no impact on simulated burnt area. Current moisture levels were the dominant influence on fuel drying. Additionally, antecedent moisture levels were important for fuel build-up. The models also enabled the visualisation of interactions between variables, such as the importance of antecedent productivity coupled with instantaneous drying. The length of the period which needs to be considered varies across biomes; fuel-limited regions are sensitive to antecedent conditions that determine fuel build-up over longer time periods ($\sim$4 months), while moisture-limited regions are more sensitive to current conditions that regulate fuel drying.

## 1 Introduction

Wildfires are an important natural disturbance of the Earth System. They have extensive socio-economic impacts as well as profound effects on vegetation, atmospheric composition, and climate (Bowman et al., 2011; Voulgarakis and Field, 2015; Andela et al., 2017; Lasslop et al., 2019). How fire regimes may change in the future, and how fire-related feedbacks may influence climate and global environmental changes are growing concerns.

The factors that influence the occurrence and intensity of fire are well-known: the presence of an ignition source, vegetation properties that determine the availability of fuel, and weather conditions that promote fuel drying and thereby the rate of fire spread. However, these factors are strongly coupled to one another. Climate conditions influence the incidence of lightning and the nature of the vegetation, while wind strength and the impact of atmospheric conditions on drying are modulated by vegetation cover. Furthermore, the relationships among ignitions, vegetation, and climate may depend on the timescales involved; short-term drought promotes fuel drying and hence increases fire risk, but in the longer term, drought conditions reduce vegetation cover and fuel loads. This complexity makes it challenging to disentangle the causes of observed changes in fire activity.

Furthermore, recent declines in burnt area (BA) in some regions have been explained as a consequence of human activity, through indirect and direct intervention (Martínez et al., 2009; Andela et al., 2017) albeit modulated by climate and vegetation (Forkel et al., 2019b). Such human intervention can promote or suppress fire through ignitions, fuel management, and landscape modification. A mainly temperature-driven increase in conditions conducive to wildfires was suggested by a number of regional studies (e.g. Westerling, 2006; van Oldenborgh et al., 2020; Goss et al., 2020; Barbero et al., 2015). At the global

scale, Abatzoglou et al. (2019) showed that anthropogenic climate change had led to an increase in fire weather over 22% of the global burnable area by 2019, while Jolly et al. (2015) found that anthropogenic climate change has led to a lengthening of the fire season across more than a quarter of global vegetated land in recent decades. Increases in fire weather are predicted under different assumptions about levels of future warming (e.g. Burton et al., 2018; Turco et al., 2018; Bedia et al., 2015).

Understanding the interplay among the different present-day controls of fire is also a key requirement for the prediction of future fire-regime shifts and impacts on the land biosphere and human activities. Coupled fire-vegetation models can be used to predict changes in large-scale fire regimes in response to future climate change scenarios (see e.g. Knorr et al., 2016; Kloster et al., 2012) and to explore how these changes are affected by and will affect regional vegetation patterns and climate. Although these models are reasonably good at simulating modern geographical fire patterns in BA, they are poor at reproducing observed fire-season length and inter-annual variability (IAV) in BA (Hantson et al., 2020). Furthermore, there are large differences in their predictions of both historical (Teckentrup et al., 2019) and future (Kloster and Lasslop, 2017; Sanderson and Fisher, 2020) trends.

Studies have pinpointed the relationship between simulated vegetation properties and BA as a cause for concern (e.g. Forkel et al., 2019a; Kelley et al., 2019; Teckentrup et al., 2019; Hantson et al., 2020). Forkel et al. (2019a) analysed satellite data to show that while state-of-the-art fire-vegetation models reproduce the emergent relationships with climatic variables, they do not correctly represent the relationship between vegetation and BA. Hantson et al. (2020) highlighted the need for improved understanding of vegetation drivers of fire season length and IAV of BA. Both Forkel et al. (2019a) and Hantson et al. (2020) argued for a better understanding of how vegetation properties control fuel build-up, and therefore fire occurrence and intensity.

Fuel is organic matter that is available for ignition (Keane et al., 2001). The type, amount, and spatial arrangement of fuel affect its tendency to burn (Archibald et al., 2009). These properties, dictated by vegetation, in turn affect fuel connectivity and hence fire spread in addition to how rapidly fuel dries out and becomes combustible. Antecedent weather conditions in the weeks to years before fire events can determine fuel availability (van Oldenborgh et al., 2020) and hence fire occurrence. The effect of antecedent weather conditions on BA may depend on the types of vegetation present (which influences whether fuel drying or accumulation is most important): antecedent precipitation will increase BA in fuel-limited regions, for example, but decrease BA in regions where fuel drying is the major control (Alvarado et al., 2020; Abatzoglou and Kolden, 2013; Littell et al., 2009).

A number of regional and global studies have indicated the importance of antecedent fuel build-up for BA. For example, links between fire activity and antecedent productivity have been found in South Africa (Van Wilgen et al., 2000), central Australia (Griffin et al., 1983), grass and shrublands in the western United States (Littell et al., 2009; Westerling et al., 2003; Swetnam and Betancourt, 1998), New South Wales, Australia, for bushfire fuel (Jenkins et al., 2020), and southern Africa (Archibald et al., 2009). Global studies have identified similar relationships (a positive relationship between pre-season productivity and fire activity in the following dry season) in some dry areas. By studying the correlation between growing period (i.e. antecedent) soil moisture and fire activity, Krawchuk and Moritz (2011) found fire activity in dry regions to be related to antecedent productivity. Similarly, van der Werf et al. (2008) found a similar relationship for arid ecosystem (e.g. northern Australia), where antecedent wet conditions coupled with instantaneous drying were found to be important. Other

global studies have also identified northern Australia as obeying this relationship (Randerson et al., 2005; Spessa et al., 2005). In a more recent global analysis, O et al. (2020) found that for arid regions, wet anomalies (soil moisture) lead to increased fire later in the year by increasing fuel loads and biomass. Thus, it is clear that a better understanding of the timescales of fuel accumulation, the interaction between biophysical drivers and fuel build-up, and the effects of antecedent weather conditions on both fuel loads and fuel drying is needed in order to improve predictions of BA.

While other studies have used machine learning to explore fire drivers including the effect of antecedent productivity (e.g. Archibald et al., 2009; Forkel et al., 2017; Joshi and Sukumar, 2021), they have not explored the relationship between antecedent conditions (fuel load and drying) and fire in detail. Here we quantify the roles that antecedent vegetation productivity and aridity play relative to instantaneous conditions, the critical number of months that are most important for each, the shape of their relationships to BA, and the interactions between them. While the (relative) importance of antecedent variables has been investigated before (Bessie and Johnson, 1995), we aim to quantify this on a global scale. Since other climate factors, ignitions, and human activities also influence BA, we necessarily include these factors in our analysis. The use of a machine learning approach enables us to identify non-linear relationships and interactions between the drivers. This is then combined with analysis and visualisation techniques that provide insights into the modelled relationships while mitigating the effects of correlations among variables. Such insights include the effect of a particular driver on BA and the interactions between pairs of drivers.

## 2 Methods

### 2.1 Data

The predictor and BA datasets are available for different but overlapping time periods (Table 1). We pre-processed each dataset separately and conducted random forest analyses based on the common period from January 2010 to April 2015. Monthly fractional BA for this period was obtained from the GFED4 dataset (Giglio et al., 2013) (data were retrieved from https://www.globalfiredata.org/data.html). A longer time period from November 2000 to December 2019 was also considered in an analysis using fewer variables.

Diurnal temperature range (DTR), maximum temperature (MaxT), dry-day period (DD), and soil moisture are important climate factors influencing BA (Archibald et al., 2009; Bistinas et al., 2014; Forkel et al., 2017, 2019a; Abatzoglou et al., 2018; Kelley et al., 2019) and are thus considered as predictors in our analyses. DTR was calculated by taking the monthly average of the difference between the daily maximum and minimum ERA5 (Copernicus Climate Change Service (C3S), 2017) 2 m temperatures.

The dry-day period was defined as the longest contiguous period of ERA5 mean daily precipitation below 0.1 mm day$^{-1}$ (wetting rainfall; Harris et al., 2014; Jolly et al., 2015) within each month. A period contiguous with the previous month's dry-day period was concatenated such that the sum of both (number of days) was used to determine the longest period. For example, consider a 30-day long month with a 10-day long dry-day period at the beginning of the month, followed by a wetting precipitation event on day 11, and then a dry-day period for the following 19 days. This month has a

dry-day period of 19 days. However, if the previous month were to terminate in a 10-day long dry-day period, these 10 days would be added to the initial 10-day dry-day period of the current month, thereby making this combined dry-day period the longest.

Soil moisture was taken from the Copernicus soil water index (SWI) dataset (Albergel et al., 2008; Wagner et al., 1999). We used the WGLC dataset (Kaplan and Lau, 2019) which provides counts of monthly lightning strikes. It is based on the World Wide Lightning Location Network (WWLLN) dataset, which mainly detects cloud-to-ground strikes (Rodger et al., 2004; Abarca et al., 2010), as opposed to LIS lightning data (Bürgesser, 2017).

Land cover was shown in previous studies to be another important influence on BA. We included several alternative representations of land cover including above-ground tree biomass (AGB) and the fractional cover of trees (TREE), shrubs (SHRUB), herbaceous vegetation (HERB), and crops (CROP) in our predictor set. Tree AGB was obtained by mosaicking AGB datasets for the tropics (Avitabile et al., 2016, 1 km resolution) and northern forests (Thurner et al., 2014, $0.01°$ resolution) using the mean after resampling each to a common spatial resolution of $0.25°$. Yearly land cover values were obtained from the ESA CCI Land Cover dataset (Li et al., 2018). Land cover types were converted to fractional cover according to Poulter et al. (2015) using the conversion table as in Forkel et al. (2017). Global population density (POPD) from an updated version of the HYDE 3.2 dataset (Klein Goldewijk, 2017, Kees Klein Goldewijk; personal communication, February 2021) was used as a measure of human influence on vegetation and fire regimes.

Field data on fuel loads is sparse and the only global dataset (Pettinari and Chuvieco, 2016) is based on extrapolating scattered field measurements by biome. We therefore used four remotely sensed vegetation properties related to total biomass or leaf cover that could be regarded as indices for fuel load in our predictor set: solar-induced fluorescence (SIF), vegetation optical depth (VOD), fraction of absorbed photosynthetically active radiation (FAPAR), and leaf area index (LAI). All four properties have previously been used as productivity indices (e.g. Mohammed et al., 2019; Ryu et al., 2019; Teubner et al., 2018; Ogutu et al., 2014) and we use all four because it is uncertain which would be most closely related to fuel loads. Monthly SIF was obtained from the GlobFluo SIF dataset (Köhler et al., 2015). Ku-band VOD was obtained from the VODCA dataset (Moesinger et al., 2020). FAPAR and LAI were obtained from the MOD15A2H dataset (Myneni et al., 2015). To pre-process data for the period from January 2010 to April 2015 we used data from January 2008 to April 2015, for which period all four datasets are available. Similarly, relevant data from February 2000 to December 2019 was pre-processed to enable analysis of the period from November 2000 to December 2019.

**Table 1.** Characteristics of the datasets. End times as applicable to the processed data are indicated in brackets.

| Variable | Abbreviation | Dataset | Start | End | Time | Reference |
|---|---|---|---|---|---|---|
| burnt area | BA | GFED4 | 06-1995 | 12-2016 | monthly | Giglio et al. (2013) |
| diurnal temperature range | DTR | ERA5 | 1950 | present (12-2020) | monthly | Copernicus Climate Change Service (C3S) (2017) |
| maximum temperature | MaxT | ERA5 | 1950 | present (12-2020) | monthly | Copernicus Climate Change Service (C3S) (2017) |
| dry-day period | DD | ERA5 | 1950 | present (11-2020) | monthly | Copernicus Climate Change Service (C3S) (2017) |
| soil moisture | SWI | Copernicus SWI | 01-2007 | 11-2018 | monthly | Albergel et al. (2008); Wagner et al. (1999) |
| lightning | Lightning | WGLC Lightning | 01-2010 | 12-2018 | monthly | Kaplan and Lau (2019) |
| above-ground tree biomass | AGB | Tropical AGB: Avitabile, Northern AGB: Thurner | static | static | static | Avitabile et al. (2016); Thurner et al. (2014) |
| land cover (fractional cover per grid cell) | CROP, SHRUB, TREE, HERB | ESA CCI Land Cover | 1992 | present (2019) | yearly | Li et al. (2018) |
| solar-induced fluorescence | SIF | GlobFluo SIF | 01-2007 | 04-2015 | monthly | Köhler et al. (2015) |
| vegetation optical depth | VOD | VODCA (Ku-band) | 12-1997 | 12-2018 | monthly | Moesinger et al. (2019) |
| fraction of absorbed photosynthetically active radiation | FAPAR | MOD15A2H | 02-2000 | present (03-2021) | monthly | Myneni et al. (2015) |
| leaf area index | LAI | MOD15A2H | 02-2000 | present (11-2018) | monthly | Myneni et al. (2015) |
| population density | POPD | HYDE 3.2 (updated) | 2000 | present (2020) | yearly | Klein Goldewijk (2017), Kees Klein Goldewijk; personal communication (February 2021) |
| MCD64CMQ burnt area | MCD64 BA | MCD64CMQ | 11-2000 | present (06-2020) | monthly | Giglio et al. (2018) |

## 2.2 Data processing

### 2.2.1 Gap filling

There are gaps in the SWI, FAPAR, LAI, SIF, and VOD datasets in winter months at latitudes above $\sim$60°N, and in the austral winter for southern South America, due to high solar zenith angles for FAPAR, LAI, and SIF and because of snow cover and frozen soil for SWI and VOD (see e.g. Moesinger et al., 2020). There are also sporadic missing values in these datasets caused by e.g. cloud cover. Unfortunately, simple exclusion of the times lacking data is not possible for our analysis because we commonly rely on antecedent samples throughout. Thus, data gaps were filled using a two-step approach as in Forkel et al.

(2017) in order to allow analysis of summer months at the affected locations. This approach differentiates between two gap types based on the amount of missing information for a specific month at each location.

First, 'persistent' gaps, defined as months for which $50\%$ or more of the observations across all years are missing, were filled using the minimum value observed at that location for the given predictor variable. We assume that this indicates missing data during the winter, since other causes for data gaps (e.g. cloud cover) are predominantly 'transient'. For example, if a certain
grid cell was missing data for more than $50\%$ of all Decembers in the record, these gaps in December would be treated as persistent and therefore filled using minima.

Second, the remaining transient gaps were filled using season-trend regression models with four harmonic terms ($k = 4$) and without breakpoints. These models were fitted using ordinary least squares regression to the entire timeseries obtained during the first step, as mentioned before using data from January 2008 to April 2015 (or November 2000 to December 2019 for
the monthly analysis). Cloud cover, which also affects detection in tropical and subtropical regions, is usually transient, and therefore filled using the regression models. Locations where no observations were available for $> 52$ months out of the total 88 months (regardless of whether such data gaps always occur in the same month, as for persistent gaps, or at any point throughout the year) were discarded in a trade-off between data quality and geographic extent. For the monthly analysis, locations were discarded given $> 138$ unavailable months (out of 239 months).

Use of a different gap filling mechanism (Fig. S1b; temporal nearest-neighbour gap-filling) yielded very similar results. This simple nearest-neighbour gap-filling approach used for the eventual ALL_NN model processes timeseries at a given location, filling gaps by using the temporally closest available samples at that location. Of the two approaches, we decided to use the season-trend model with minima filling because it represents a more physical solution; it is based on an approach previously used for vegetation variables (see Beck et al., 2006), for which one would expect minima to occur during winter. Indeed, as
can be seen in Fig. S2, virtually no samples are being filled with minima outside of winter, and predominantly in the northern extreme latitudes. While our gap-filling methodology may yield unphysical values for non-vegetation variables like SWI, we do not expect the filling of SWI to have a big influence on the final results because we do not use antecedent values of SWI. Since we do not anticipate fires during the winter, having (by necessity of gap filling) potentially unphysical values of SWI in the winter should not affect results where relevant for our analysis.

### 2.2.2 Interpolation

All datasets were interpolated to a common $0.25°$ spatial grid. Datasets where the original spatial resolution was higher than this were averaged; the other datasets were interpolated using nearest-neighbour interpolation to avoid smoothing local extrema (Forkel et al., 2017). Datasets that were only available at yearly time resolution (i.e. land cover, POPD) were linearly interpolated to monthly intervals. Temporally static data (i.e. AGB) were recycled. Processing was carried out before averaging
to provide monthly climatological time series where applicable.

### 2.2.3 Antecedent predictor variables

The influence of antecedent conditions that might affect fuel loads or fuel dryness, specifically vegetation properties and DD, on BA was investigated by using antecedent FAPAR, LAI, VOD, SIF, and DD data from up to two years before any given month (1M, 3M, 6M, 9M, 12M, 18M, 24M, where M denotes months). The large autocorrelation between predictor variables could impede the visual interpretation of the impacts of antecedent periods $\geq 1$ yr. Thus, anomalies were computed by subtracting the seasonal cycle relative to the designated month, resulting in the following transformations:

$$(\text{X } 12\text{M}) - (\text{X } 0\text{M}) \rightarrow \text{X } \Delta 12\text{M},$$

$$(\text{X } 18\text{M}) - (\text{X } 6\text{M}) \rightarrow \text{X } \Delta 18\text{M},$$

$$\text{and } (\text{X } 24\text{M}) - (\text{X } 0\text{M}) \rightarrow \text{X } \Delta 24\text{M},$$

where $\text{X} \in \{\text{FAPAR, LAI, VOD, SIF, DD}\}$ and X 0M refers to the variable X in the current month. For example, the 12-month antecedent X 12M was transformed by subtracting the instantaneous (month 0) value of X, thereby yielding the anomaly in X, X $\Delta 12\text{M}$, that may be easier to interpret.

### 2.3 Machine learning experiments

We used random forest (RF) regression to model the relationships between BA and the driver variables (predictors). RF is an ensemble learning approach in which multiple decision trees are constructed using a randomly sampled subset of training observations. The final model is the average result from all of the individual decision trees. RF regression is highly suited to investigating the emergent controls on fire because it is able to learn non-linear relationships in high-dimensional space (Archibald et al., 2009). By averaging over multiple decision trees, RFs also mitigate overfitting (Breiman, 2001). We used the scikit-learn version 0.24.1 (Pedregosa et al., 2011) RF regression implementation in Python, with hyperparameters determined using five-fold random cross validation (CV) of the eventual ALL model: `n_estimators`: 500, `max_depth`: 18, and default values for all other parameters. The number of estimators (`n_estimators`) determines the number of trees whose predictions are averaged. The maximum depth (`max_depth`) limits the number of split levels, which can reduce overfitting. We also found that a limited number of split levels was necessary for the computation of SHAP values, although we expect this to be a limitation of the specific software we used as opposed to the SHAP method itself. The hyperparameters were only estimated once for the model containing all variables due to computational constraints.

The validation dataset was randomly sampled across space and time and comprised $30\%$ of the data. To estimate how the model will perform on unseen data, the out-of-bag (OOB) $R^2$ for the training dataset can be used (Fox et al., 2017). However, since this data still belongs to the training dataset, the $R^2$ for the validation dataset, which has not been used for variable selection or hyperparameter tuning, is also used to provide an alternative, independent, measure of the generalisability of a given model.

However, this is only valid if there is no autocorrelation between the samples. We investigated the degree of spatial autocorrelation using a variogram of global GFED4 BA which informed a buffered leave-one-out (B-LOO) CV procedure follow-

ing Ploton et al. (2020). This was carried out to determine how much the autocorrelation that may be present influences the amount of potential overfitting. We did not employ the extrapolation-prevention procedure used in Ploton et al. (2020) because it led to the exclusion of significant areas like northern Australia and west Africa. The B-LOO CV was executed as follows, where $r_{max} = 50$ pixels and $N_t$ was chosen such that the number of potential training samples was guaranteed to be equal to or above $N_t$ for all $r \leq r_{max}$:

1. Randomly choose a single location. The 12 monthly samples at this test location constitute the test set.

2. Exclude samples from the potential training set in a circular region of radius $r$ pixels around the test location, such that no potential training sample is closer than $r$ pixels to the test location. This limits the influence of spatial autocorrelation.

3. Randomly choose $N_t$ training samples from all remaining potential training samples.

4. Using a model trained on the above training samples, predict BA for the test location.

5. Increment $r$ and repeat steps 1–4 until $r$ has reached $r_{max}$.

This process was repeated 4000 times for each of the eight linearly spaced investigated radii, with the lowest radius being equal to 0. Due to computational constraints, the B-LOO CV was only carried out for a single set of variables.

We trained a number of different RF regression models to test explicit hypotheses about the importance of antecedent conditions on BA (see Table 2) using the defined hyperparameters on the climatological timeseries. The initial experiment (ALL) was run using the basic set of 15 predictor variables related to climate, vegetation, and human influences on fire (Table 1) and included both current and antecedent values of the four vegetation indices and DD, giving 50 predictor variables. A second experiment (TOP15) used only the 15 most important predictors from the ALL model, as a way of testing whether all the predictors were necessary and whether including so many predictors resulted in overfitting.

The choice of 15 predictors was heuristically based on the slope of the feature importance plots (see Fig. S3), where, by inspection, the importance change is minimal after 15 variables. Thereafter, no additional information was being conveyed, so we decided to use this as our threshold. While use of the more rigorous recursive feature elimination with cross-validation (RFECV) would be possible in principle, this commonly makes use of the Gini importance owing to its ease of calculation, as it only considers data already seen during training. Unfortunately, this also means that RFECV fails to account for overfitting, as it only considers the training data when calculating feature importance (Meyer et al., 2019). In contrast, the different approaches we jointly utilised to calculate a more robust feature importance metric are much more computationally demanding, making RFECV infeasible.

All of the remaining experiments used combinations of 15 predictor variables. The CURR experiment only used current-month values of each predictor. Therefore, comparison of the CURR and ALL experiments allowed the impact of including antecedent vegetation and moisture conditions to be evaluated. However, some of the vegetation predictors are highly correlated with one another which could artificially decrease their importance. To test this, we ran four further experiments (15VEG_FAPAR, 15VEG_LAI, 15VEG_VOD, 15VEG_SIF) that included the 10 most important non-vegetation predictors

from the ALL model, potentially including current and antecedent values of DD. In addition, each of these experiments con-
tained one of the four vegetation predictors represented by both current (0M) and antecedent values (1M, 3M, 6M, and 9M). To
disentangle the effects of antecedent DD and antecedent vegetation properties, we ran a second set of vegetation experiments
(CURRDD_FAPAR, CURRDD_LAI, CURRDD_VOD, CURRDD_SIF) where each vegetation predictor was represented by
both current (0M) and antecedent values (1M, 3M, 6M, and 9M) but only using current DD and the next nine most important
non-vegetation factors from the CURR model. Finally, five-fold random CV was used to isolate the best combination of the
vegetation predictors under the constraint that each of the five states (0M–9M) must be represented exactly once (using any of
the four vegetation predictors), resulting in the BEST15 model.

In addition to the above climatological experiments, we also investigate monthly data for the time period 11-2000–12-2029
(230 months) using the 15VEG_FAPAR_MON model. To avoid the temporal limits of the GFED4 dataset (see Table 1) the
MODIS MCD64CMQ (Giglio et al., 2018) BA dataset was used. Otherwise, this experiment uses the same variables as the
15VEG_FAPAR experiment with the exception of lightning, which was replaced with the similarly significant variable AGB
(see Fig. S3) in order to enable processing of a longer time period. In addition to five-fold random CV as for the other models,
the performance and generalisability of this model was also measured using temporal CV. Here, the model was trained on
all samples excluding either all months within the years 2009–2012 (including 2012) or 2016–2019 (including 2019). There-
after, the $R^2$ was measured on whichever years were excluded for training. Note that unless explicitly specified, all following
methodological descriptions will relate to the climatological experiments as opposed to the monthly 15VEG_FAPAR_MON
experiment.

### 2.4   Measuring predictor importance and relationships

Our goal is to determine the contribution of individual predictors (including antecedent states of these predictors) to model
skill at predicting BA and to examine the relationships between predictors and BA. There is no unique way to measure the
importance of a given predictor on model skill in predicting BA and it is particularly difficult to assign importance to individual
predictors when there is a high degree of collinearity between them (Dormann et al., 2013; Nowack et al., 2018; Mansfield et al.,
2020). We use four techniques to infer the importance of individual predictors: Gini impurity, permutation feature importance
(PFI), leave-one-column-out (LOCO), and SHapley Additive exPlanations (SHAP) values. The Gini importance aggregates
the decrease in mean squared error (MSE) for each split involving a given predictor variable over the individual decision trees
making up the RF. The PFI was calculated from five permutations of each predictor variable (validation set) using the ELI5
0.11.0 PermutationImportance (https://eli5.readthedocs.io/en/latest/index.html). While this provides an alternative assessment
of the prediction score, the permutations may result in unlikely or impossible combinations of predictors and thus the PFI
approach has a known tendency to overemphasise the importance of individual variables (Hooker and Mentch, 2019). The
LOCO importance measure is estimated by repeatedly retraining the RF models, each time without one particular predictor
variable. The relative importance of this predictor variable is then measured as the change in MSE on the validation dataset,
where a larger drop in MSE signifies a larger significance for the variable within the dataset. The importance of correlated
predictor variables may be under-emphasised in this approach since the model is retrained, and thus some of the importance

**Table 2.** The modelling experiments. Except for the ALL experiments, the other experiments included 15 predictor variables for comparability. Differences in number of antecedent variables included in each experiment meant that different numbers of variables from the basic set were used in these experiments. For the TOP15, 15VEG_X(_MON), and BEST15 models we used the most important variables from the ALL experiment up to the required number of 15. For the CURRDD_X models, we used the most important non-vegetation variables from the CURR model. Table S1 provides a detailed list of the variables included in each experiment.

| Name | Nr. | Variables |
|------|-----|-----------|
| ALL & ALL_NN | 50 | Basic set of current variables + current month and antecedent (1, 3, 6, 12, 18, 24M) values for Dry Days and vegetation indices (FAPAR, LAI, VOD, SIF) |
| TOP15 | 15 | Top 15 predictors from the ALL model |
| CURR | 15 | Only current values of the basic set of 15 variables |
| 15VEG_X (e.g. 15VEG_FAPAR) | 15 | Top 10 non-vegetation variables from the ALL experiment, plus current and antecedent (1, 3, 6, 9 months) vegetation index X (e.g. FAPAR) |
| CURRDD_X (e.g. CURRDD_FAPAR ) | 15 | Current and antecedent (1, 3, 6, 9 months) versions of the vegetation index X (e.g. FAPAR), current DD, top 9 other variables from the CURR experiment |
| BEST15 | 15 | Current and antecedent DD, one current, 1M, 3M, 6M, 9M vegetation index (drawn from the four potential vegetation indices) and 5 most important other variables from the basic set |
| 15VEG_FAPAR_MON | 15 | Same as the 15VEG_FAPAR experiment with monthly data instead of climatological data and lightning data replaced by the next-most important non-vegetation variable |

associated with the removed variable may be transferred to the correlated variables during the re-training process. The SHAP value (Lundberg and Lee, 2017; Lundberg et al., 2020) is the average of the marginal contributions from a series of perturbations of the predictor variables. In a similar way to the PFI, this method shares the importance amongst correlated predictor variables, which may make them appear less significant than if they were included on their own. SHAP values were computed for all validation samples. In order to create a robust composite importance metric for each predictor variable, we divided the Gini, PFI, LOCO, and SHAP metrics by the sum of their absolute values and then summed them.

Maximally significant timescales were calculated by weighting antecedent months using the largest SHAP value magnitude out of all 12 months in the climatological data. The maximum SHAP value magnitudes were calculated for a predictor variable $x$ at location $\ell$ on the latitude-longitude grid as follows:

$$\text{SHAP}_{x,\ell} := \text{SHAP}_{x,\ell,m_{\max}}, \quad \text{where } m_{\max} = \underset{m \in \{1,2,...,12\}}{\arg\max} |\text{SHAP}_{x,\ell,m}|. \tag{1}$$

These were then used to calculate the maximally significant timescales for the basis predictor variable X (e.g. FAPAR) using an average over antecedent months, a, weighted by maximum SHAP value magnitudes:

$$t_{\max,X,\ell} = \frac{\sum_{a,x} a|\text{SHAP}_{x,\ell}|}{\sum_{a,x} |\text{SHAP}_{x,\ell}|} \quad \text{for } a, x \in \{(0, X\,0M), (1, X\,1M), (3, X\,3M), (6, X\,6M), (9, X\,9M)\}. \tag{2}$$

Locations with too many significant antecedent months were ignored in order to visualise resulting relationships more reliably; for example, if both the current ($|\text{SHAP}_{X\,0M,\ell}|$) and nine-month antecedent ($|\text{SHAP}_{X\,9M,\ell}|$) magnitudes are dominant,

the weighted mean month (according to Eq. (2)) would lie in between, which is physically meaningless. We designed an algorithm to detect SHAP values that differ significantly from the baseline in order to mitigate this. Additionally, we also applied a range-based threshold, whereby locations were ignored if the variability of the SHAP values at location $\ell$ was below a threshold heuristically related to the mean BA, $\overline{\text{BA}}_\ell$ (based on all BA samples):

$$\max_x(\text{SHAP}_{x,\ell}) - \min_x(\text{SHAP}_{x,\ell}) < 0.12 \times \overline{\text{BA}}_\ell. \tag{3}$$

We further used Accumulated Local Effects (ALE) plots (Apley and Zhu, 2020) to examine and interpret the coupled relationships fitted by the RF models. ALE plots are a more robust alternative to partial dependence plots (PDP) or individual conditional expectation (ICE) plots (Apley and Zhu, 2020; Molnar, 2020). We assessed the impact of each of the predictor variables on BA in isolation using first-order ALEs which take into account the effect of all other predictor variables. However, underlying inhomogeneities may appear when the model fits different relationships for different locations or times. We therefore tested for inhomogeneities by subsampling the dataset prior to ALE plotting to enable the visualisation of underlying relationships for a subset of locations and times. The causes of these inhomogeneities were explored using second-order ALE plots, which show the combined effect of two predictor variables on BA.

## 3 Results and discussion

In general, all models are able to predict BA using the given biophysical predictors. However, the inclusion of antecedent predictors significantly improves model performance. Below, we discuss the performance of the different models, the importance of the different predictor variables, and their relationships with BA.

### 3.1 Model performance

The ALL model, which includes all 50 variables, achieves an in-the-bag $R^2$ of 0.919 and out-of-bag (OOB) $R^2$ of 0.701 for the training dataset and an $R^2$ of 0.701 for the validation dataset (Fig. 1). Predictions on the validation set (Fig. 2b) show a similar geographic pattern to observed BA (Fig. 2a). However, overprediction in the validation set relative to observed BA is more widespread than underprediction (Fig. 2c). Nonetheless, there is no bias; the ALL model predicts a mean out-of-sample BA of $2.54 \times 10^{-3}$ compared to the expected $2.48 \times 10^{-3}$. The apparent overprediction is the result of plotting relative (as opposed to absolute) errors, which amplifies the fact that the ALL model does not predict very low BA accurately; out-of-sample BA predictions are no lower than $7.39 \times 10^{-7}$, while the observed BA is 0 for $85.7\%$ of samples. Generally, the model captures intermediate BA better than extreme BA, leading to overprediction at low and underprediction at high BA values. Thus, more samples are over-predicted because there are more values with low BA than high BA, leading to many instances of slight overprediction balanced by few instances of comparatively large underprediction (Fig. S4, S5). The model may struggle to predict 0 BA because the random forest model consists of many smaller decision trees. All 500 individual models would have to predict 0 to yield this value overall, which does not appear to occur given the stochastic nature of the training process. Failure

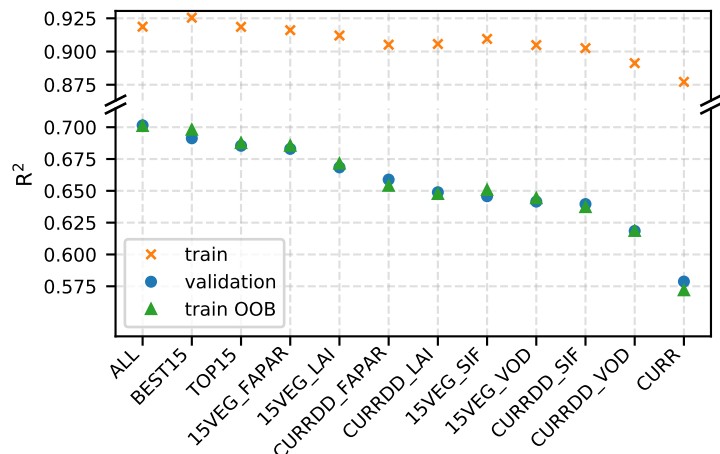

**Figure 1.** Global climatological $R^2$ scores for the different experiments. The CURR model is the only model that does not include antecedent conditions, and it performs much worse as a result. Despite the fact that the ALL model contains 50 predictors, while all other models contain just 15, it does not perform significantly better than the best models containing just 15 predictors (e.g. 15VEG_FAPAR). Note that although train $R^2$ scores are shown here, they are not indicative of model performance on unseen data, for which the shown train OOB scores should be used instead.

to capture extreme events well is likely due to their rarity, resulting in the absence of comparable training data (see also e.g.
Joshi and Sukumar, 2021).

    Using a combination of regional neural networks trained on fewer variables at a coarser spatial resolution of $1° \times 1°$, Joshi and Sukumar (2021) found a global $R^2$ score for BA prediction of $0.36$. An earlier study by Thomas et al. (2014) considered an $R^2$ score of $0.6$ as indicating a robust prediction. Our results compare favourably to both. To further ensure model robustness, we also compared the PFI importances computed separately on the training and validation sets in Fig. S6. There is no appreciable
difference between the two, which is indicative of a lack of overfitting, since the model training has not unduly prioritised certain variables based on the training set (Dankers and Pfisterer, 2020). Additionally, using the variogram shown in Fig. S7, we carried out the B-LOO CV as detailed in Sect. 2.3 in order to investigate the influence of spatial autocorrelation on the 15VEG_FAPAR model (see Fig. S8). The performance of the model drops as a larger region around the test samples is excluded (with 30 pixels corresponding to $\sim$900 km at the equator, which is the scale of autocorrelation identified using Fig. S7).
However, as opposed to the case study in Ploton et al. (2020), the $R^2$ score plateaus at around $0.1$–$0.4$ beyond $\sim$900 km instead of dropping to $0$, thereby indicating the robustness of our model. Certain regions and extreme events are poorly captured by the model, accounting for the lower end of this range. Furthermore, the model is potentially forced to extrapolate to a larger extent as the exclusion radius is increased, leading to an overly pessimistic performance estimate. The extrapolation-prevention procedure in Ploton et al. (2020) was not used here because it led to the exclusion of certain key regions.

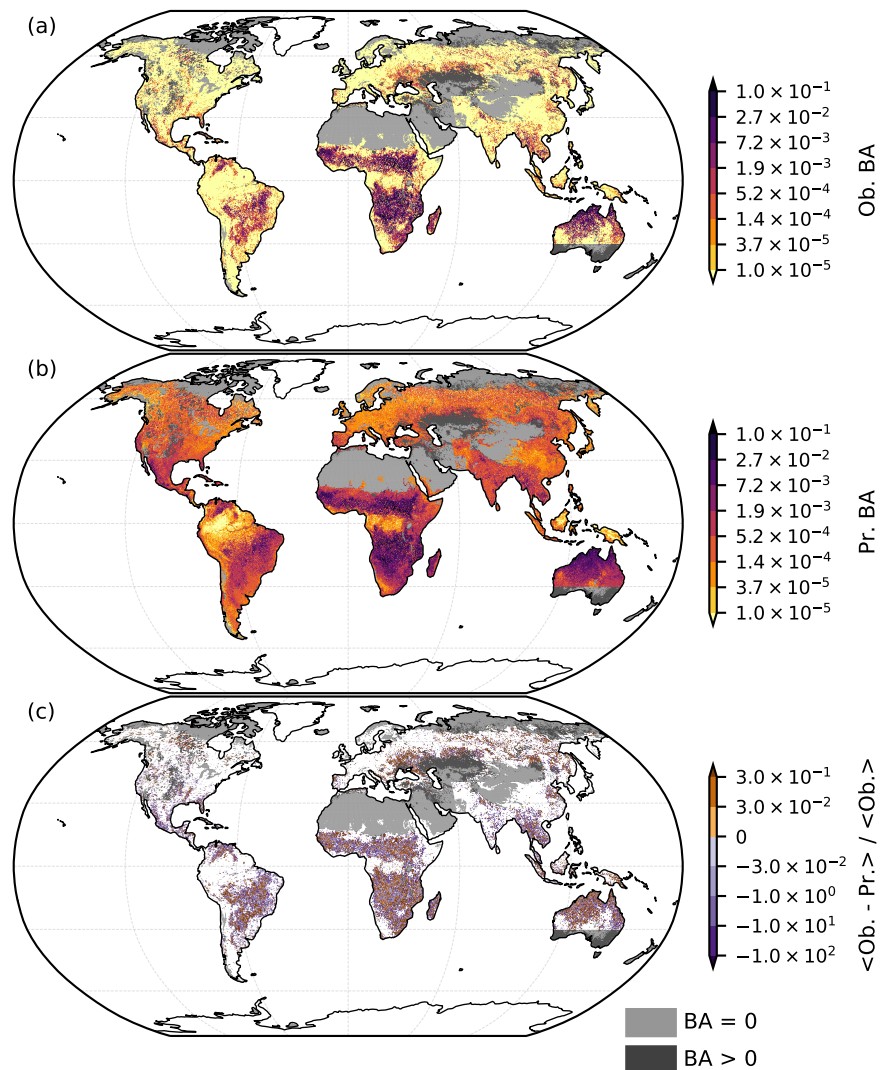

**Figure 2.** (a) Average observed (Ob.) BA derived from the GFED4 BA dataset (Giglio et al., 2013). (b) Out-of-sample predictions (Pr.) by the ALL model on the validation dataset. (c) Relative prediction error of the ALL model calculated by taking the mean of the difference between observations and predictions divided by the mean observations. While the predictions in (b) are qualitatively very similar to the observations in (a), there is an overestimation of low BA. Areas with very low or 0 observed BA (a) are omitted from (c) to avoid division by (nearly) 0. Despite the visual exaggeration of the errors, which are generally small, there is no overall pattern. Note that sharp data availability boundaries (e.g. in western Asia, southern Australia) are introduced by the AGB dataset. Grey shading indicates regions with fire data availability, but where one or more of the other datasets is not available. Light grey indicates regions where mean BA is 0, with dark grey representing regions with non-zero mean BA.

## 3.2 Importance of predictors

Climate variables and fuel-related vegetation indices have the strongest influence on BA in the ALL model (Table S2). Both current and antecedent conditions are important. Current DD and MaxT are ranked first and fifth respectively, but antecedent DD also has a moderate influence (DD 1M and DD 3M are ranked 10th and 13th in importance, respectively). Similarly, although current FAPAR is the most important vegetation index (2nd), with both current SIF and VOD occurring in the top 15 predictors, antecedent vegetation state also has a strong influence on BA. However, antecedent conditions > 9M are unimportant in the ALL model. Vegetation characteristics such as the cover of specific plant types (TREE, SHRUB, HERB) and AGB are only moderately important in determining BA (all ranked below the top 15 predictors). Human impacts, as represented by CROP and POPD, are also only moderately important globally for BA, ranked respectively 8th and 15th. Natural ignitions as represented by lightning are only ranked 21st, suggesting that at a global climatological scale burning is not limited by lightning.

The finding that fuel build-up on timescales longer than a year is not an important predictor of BA may initially be surprising given that fuel build-up as a result of fire suppression has been linked to large and catastrophic fires (e.g. in the USA; Marlon et al., 2012; Parks et al., 2015; Higuera et al., 2015). The failure to detect an influence of longer-term fuel build-up on BA probably reflects the short time interval (1–2 yrs) considered for antecedent fuel build-up, far shorter than the timescales of coarse fuel build-up in these ecosystems. The seasonal differences captured by our analyses may also be unimportant in regions where long fire return times (or fire suppression) allow fuel build-up over longer periods. Wetter forests with long fire-return intervals may also be more affected by longer term moisture deficits (Abatzoglou and Kolden, 2013) that are not captured in the limited time period analysed. However, van der Werf et al. (2008) used 13 months of fuel accumulation before the peak of the fire season to investigate herbaceous fuels, which supports our findings somewhat. It would be worthwhile to re-examine the influence of longer timescales on BA when longer datasets are available, as, even when considering the ∼20-year long MODIS record (which we do using the 15VEG_FAPAR_MON model), we are strongly limited by the data available to us. Predictability in boreal ecosystems is expected to remain very limited because the return times are many times longer than the time series, so there is a very large stochastic component. The lower performance of our monthly 15VEG_FAPAR_MON model with over 19 years of data, presented further below, in addition to the limited predictability of boreal regions found by Joshi and Sukumar (2021) despite their use of 14 years of data, both support this.

Our analyses are also impacted by the influence of previous fires on current vegetation conditions. Burnt grid cells could have a lower FAPAR, for example, as a result of prior burning within the current month. This is a problem because we are solely interested in how pre-fire vegetation conditions affect BA. The temporal and spatial scales of the analysis are responsible for this: a monthly analysis cannot resolve processes that occur on the order of days. Further, the impact of previous fires on spatially averaged vegetation properties is expected to be proportional to the burnt fraction of the affected grid cell. In savannah regions of Africa and northern Australia, where on the order of $10\%$ of a $0.25°$ grid cell may burn in a given month, this could have a significant effect on the averaged values of vegetation properties used in our analyses. Analysis using a finer spatial scale would counteract this spatial smoothing by allowing burnt pixels to be ignored so that predictor values may be estimated

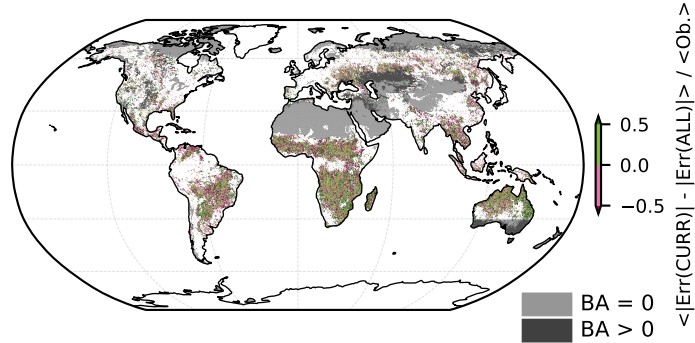

**Figure 3.** Mean change in out-of-sample prediction error between the CURR and ALL models, relative to mean observations (<Ob.>). Green regions have decreased prediction error using the ALL model compared to the CURR model, and vice versa for the purple regions. Areas with high BA (see Fig. 2a) tend to experience lower changes in relative prediction error. Areas with very low or 0 observed BA (see Fig. 2a) are omitted to avoid division by (nearly) 0. Note that sharp data availability boundaries (e.g. in western Asia, southern Australia) are introduced by the AGB dataset. Grey shading indicates regions with fire data availability, but where one or more of the other datasets is not available. Light grey indicates regions where mean BA is 0, with dark grey representing regions with non-zero mean BA.

only from unburnt cells. Using a finer temporal resolution would allow the calculation of predictor variables only up to the
time of burning. In practice, however, while many variables (e.g. MODIS-based vegetation variables) are available at finer resolutions, the lack of accurate, reliable fire statistics at finer scales (Abatzoglou and Kolden, 2013) limits the temporal and spatial resolution that can usefully be achieved.

    Although the limitation of the spatial (and temporal) resolution of the observations could impact the realism of our models, as could the omission of variables that affect fuel build-up, the consistency of the vegetation relationships shown by all the
models (as detailed below) including antecedent conditions indicates that processes related to fuel build-up are adequately represented by the chosen set of predictors. The different importance metrics used are also in broad agreement, especially regarding the most important predictors like FAPAR and DD (see Fig. S1).

### 3.3 Models with fewer predictors

The model using the top 15 predictors from the ALL model (TOP15) performs only marginally worse than the ALL model,
with an in-the-bag $R^2$ of $0.919$ and out-of-bag (OOB) $R^2$ of $0.688$ for the training dataset and an $R^2$ of $0.685$ for the validation dataset. This nearly equivalent performance reflects the fact that there is a high degree of correlation between several of the variables (Fig. S9) included in the ALL model, while also implying that the inclusion of extra predictors in the ALL model does not improve predictive capability. Therefore, this shows that it is not necessary to include multiple fuel-related vegetation variables in order to predict BA, provided that both current and antecedent conditions are taken into consideration. The removal
of predictor variables is however likely to reduce overfitting in the TOP15 model compared to the ALL model (Runge et al., 2019; Nowack et al., 2020; Joshi and Sukumar, 2021).

### 3.3.1 Importance of antecedent fuel-related predictors

The importance of antecedent fuel-related vegetation indices for predicting BA is corroborated by the results from the model that only includes predictors for the current month (CURR), where there is a large decrease in the $R^2$ for the validation dataset compared to either the ALL ($-0.123$) or TOP15 ($-0.107$) model. The decrease in the $R^2$ for the training dataset is smaller ($-0.042$) than for the validation dataset, indicating that overfitting may be more of a problem in the CURR model than the ALL model. Analysis of the mean out-of-sample prediction error shows that $54.7\%$ of grid cells are better predicted in the ALL model compared to the CURR model (Fig. 3). The performance improvements (from the CURR to the ALL model) also tend to have a larger magnitude than the performance decreases, contributing to the improvement in the global $R^2$ score. Compared to the ALL model, fuel-related vegetation properties are less important in the CURR model: VOD is the highest-ranked vegetation variable but is only fourth in importance (Table S2).

The four fuel-related vegetation variables included in the TOP15 model are correlated with one another (Fig. S9), especially on specific antecedent timescales. This suggests it may be unnecessary to include all these variables to capture the influence of fuel build-up on BA. Comparison of the models which only include current and antecedent conditions for one fuel-related vegetation variable (15VEG_FAPAR, 15VEG_LAI, 15VEG_VOD, 15VEG_SIF) confirms this. However, while all these models perform better than the CURR model (Fig. 1), only the 15VEG_FAPAR model performs similarly to the TOP15, BEST15, and ALL models. Thus, considered on its own, FAPAR is the best fuel-related vegetation predictor, followed by LAI, SIF, and then VOD (Fig. 1). However, all four fuel-related vegetation predictors produce reasonable results, and other predictors (e.g. VOD) have been found to be important in other studies (e.g. Forkel et al., 2017).

The importance of including antecedent DD is borne out by the comparison of these four experiments and the experiments which only included current DD (CURRDD_FAPAR, CURRDD_LAI, CURRDD_VOD, CURRDD_SIF). In each case, the predictions for the same vegetation predictor variable are worse (Fig. 1).

The BEST15 model contains the best combination of the fuel-related vegetation predictors (current FAPAR, FAPAR 1M, LAI 3M, SIF 6M, SIF 9M), determined by optimising their timescales. This suggests that FAPAR is most important on short timescales (current, 1M) with the other vegetation properties appearing to be more useful on longer timeframes. The performance of this model, with a training $R^2$ of $0.925$ and a validation $R^2$ of $0.691$, is only bettered by the ALL model. The good performance of models including FAPAR is due to the fact that the parts of the world responding most strongly to FAPAR 0M and FAPAR 1M tend to be fuel-limited, dry biomes accounting for the majority of global BA (Giglio et al., 2013). Therefore, globally averaged model performance metrics will tend to favour predictor variables which best represent these dominant fire regimes. This is supported by previous analyses which have found predictability in regions with infrequent fires like boreal regions or Europe to be poor in contrast to regions with more frequent fires (e.g. Joshi and Sukumar, 2021).

### 3.4 Current and antecedent relationships with BA

Current and antecedent states of both fuel-related vegetation properties and DD have different impacts on BA (Fig. 4). Current FAPAR has a negative effect on BA, while antecedent FAPAR has a positive effect on BA (Fig. 4a). The importance of current

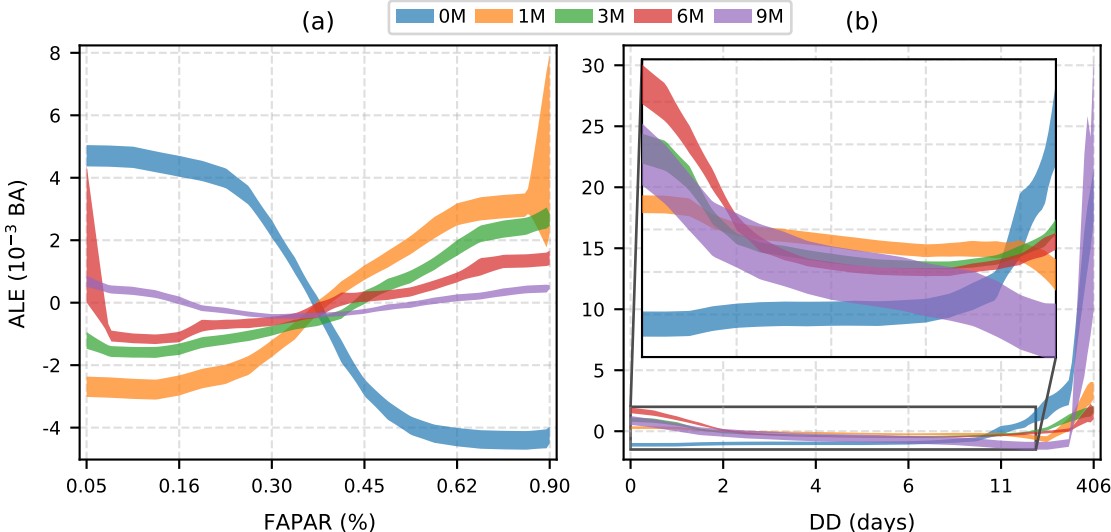

**Figure 4.** First-order ALEs for different antecedent ($< 1$ yr) relationships with (a) FAPAR and (b) DD in the 15VEG_FAPAR model, showing the underlying relationships with BA after accounting for all other variables. The shaded regions represent the standard deviation around the mean of 100 ALEs each using 122567 random samples of the training data ($\sim$10%). Evenly spaced quantiles were used in the construction of the plots. Labels were calculated using the averaged quantiles of all the variables used. A clear difference between instantaneous and antecedent relationships can be seen in both cases, with instantaneous FAPAR limiting BA while antecedent FAPAR promotes BA, and vice versa for the dry-day period. Note that the enhancement of BA due to extreme droughts (extreme dry-day period) is apparent across time periods.

FAPAR changes most rapidly at intermediate levels of FAPAR. The impact of antecedent FAPAR is strongest for the preceding 1 month but persists for up to 6 months; longer lags tend to produce results more similar to the current relationship because of autocorrelation at the yearly scale. These relationships make intuitive sense: whereas high antecedent levels of FAPAR suggest that fuel availability is not a limiting factor, high FAPAR in the current month indicates that the vegetation has sufficient moisture to be actively growing and is therefore less likely to burn.

Current DD has a positive effect on BA (Fig. 4b), while antecedent DD has a generally negative effect except if DD is very high when the effect becomes positive again. Whereas the positive antecedent effect of FAPAR on BA is strongest for the preceding 1M relationship and then gets weaker, the negative impact of antecedent DD becomes gradually stronger up to 9M. In contrast to FAPAR, dry conditions in the current month promote fire whereas dry conditions in preceding months reduce vegetation growth and hence fuel build-up. Although prolonged droughts might be expected to reduce the availability

of fuel, the (on average) positive relationship between BA and DD at very high levels of DD across all antecedent states does not support this expectation. The positive impact of drought in the current month becomes apparent for dry days $> \sim$10 days, whereas the threshold is higher for antecedent months: BA only increases when the number of dry days is $> \sim$20 days for the

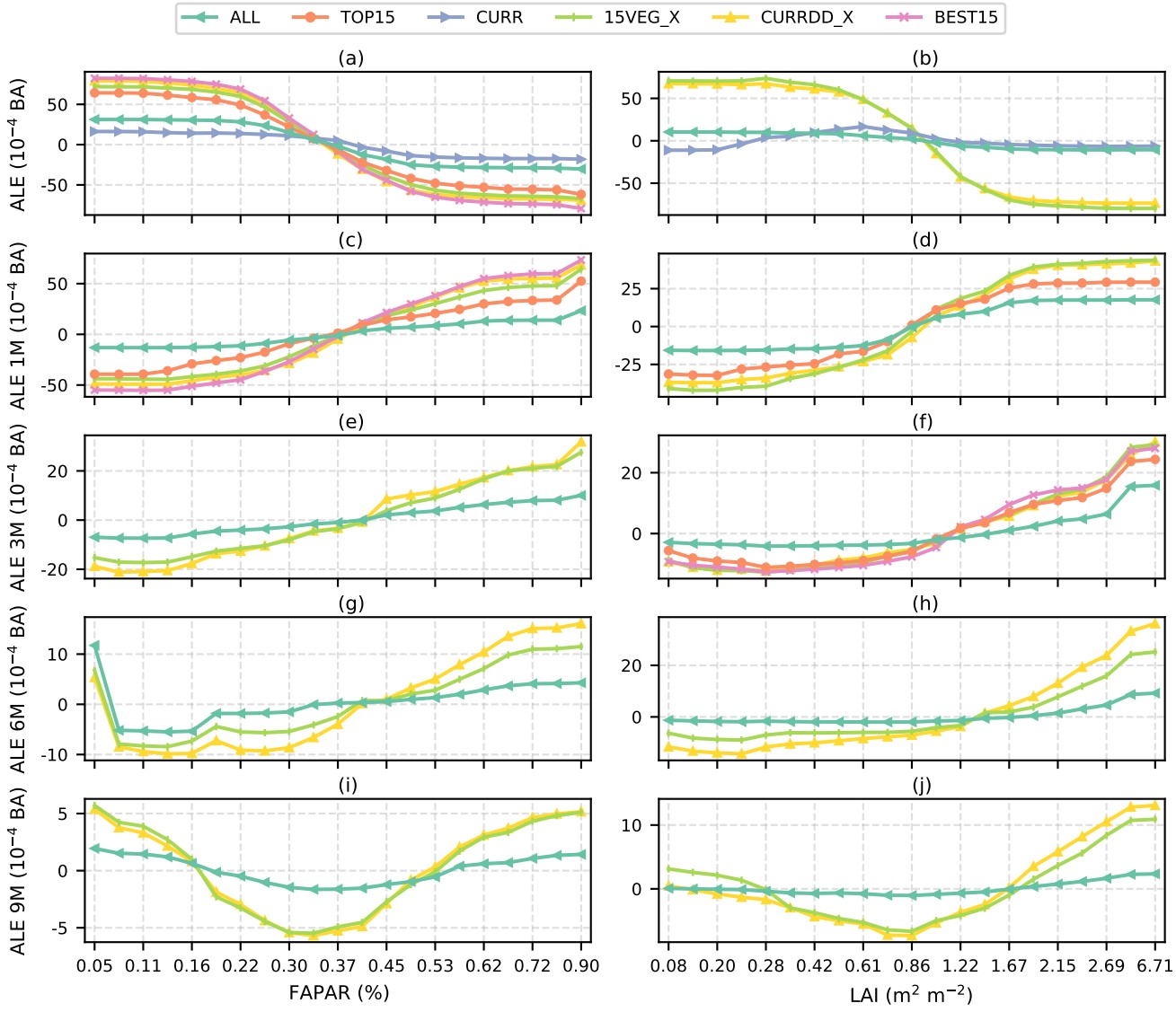

**Figure 5.** First-order ALEs for different lags ($< 1$ yr) from all relevant modelling experiments for the relationships between BA and FAPAR (left hand column) and LAI (right hand column). Evenly spaced quantiles were used in the construction of the plots. Notably, the relationship between LAI and BA is not modelled consistently by the CURR model (b), but relationships with BA are generally consistent across models otherwise.

preceding month (DD 1M) and requires $> \sim 40$ days for DD 9M. This suggests that positive large antecedent DD may reflect prolonged droughts extending into the current month, thereby increasing fuel flammability and promoting fire.

From Fig. 7a it can be seen that the negative effects of current FAPAR are most important in wet biomes, where limitation of fire activity due to instantaneous moisture conditions would also be expected to be strongest. This panel also has strong similarities with the results of Boer et al. (2021), with moisture-limited (dryness-limited) regions they identified corresponding broadly to the regions where instantaneous FAPAR is dominant due to the limitation imposed by high FAPAR on BA (see Fig. S10a). In these regions, instantaneous conditions reduce fuel available to burn due to moisture and antecedent conditions

are less important due to the lack of a seasonal fuel build-up pattern. This then shows that in moisture-limited regions, dry events are important for enabling fire. From Fig. 7b it is apparent that on average, antecedent FAPAR is most important on a $\sim$4 month timescale.

### 3.4.1  Consistency of relationships

Consistent relationships between current or antecedent conditions and BA are generally reproduced in all of the RF models

(Fig. 4, 5, S11). However, exclusion of antecedent vegetation predictors can lead to counter-intuitive relationships between the current vegetation state and BA. Although the CURR model produces the expected relationship with current FAPAR (and SIF; Fig. 5a, S11a), the relationship between current LAI (and VOD) and BA is initially positive and then flat (Fig. 5b, S11b); this model does not show the expected strong negative relationship between current LAI and BA that occurs when antecedent moisture and vegetation conditions are included.

Relationships between predictors and BA were also stable when considering the 15VEG_FAPAR_MON model (Fig. S12) which not only uses monthly instead of climatological data, but also a different BA dataset. Using random CV, a validation $R^2$ of 0.501 and an OOB train $R^2$ of 0.498 were measured. Excluding the years 2009–2012, a validation $R^2$ of 0.403 and an OOB train $R^2$ of 0.507 were measured. Excluding the final years 2016–2019, a validation $R^2$ of 0.435 and an OOB train $R^2$ of 0.505 were measured. While these $R^2$ scores are lower than those observed for the previously discussed climatological

analyses, they demonstrate that the model is able to robustly predict BA under multiple CV scenarios. Lower $R^2$ scores are also expected given the higher variance of this data. Additionally, the relationships identified by the model are highly consistent with the previous climatological analyses, showing that there is no temporal change that is important. The spatial patterns are dominant as the models behave very similarly when fit on climatological and monthly data; and the main commonality between those data is the geographical pattern. Note also that while lightning is omitted from this experiment in contrast to the

climatological 15VEG_FAPAR experiment, lightning is also not present in the TOP15 model which performs similarly to the ALL and BEST15 models. Furthermore, as shown in Fig. S3, the importance of lightning and its replacement, AGB, are very similar.

### 3.4.2  Interactions

Although it is informative to consider the impact of individual predictor variables on BA, the expression of these relationships

in the real world is likely to be conditioned by interactions with other variables. For example, low values of current FAPAR are associated with high BA (Fig. 4a), but this association occurs only when antecedent FAPAR is high (Fig. 6). Low FAPAR in the current month reflects unsuitable conditions for plant growth, for example during the dry season. Therefore, fuel build-

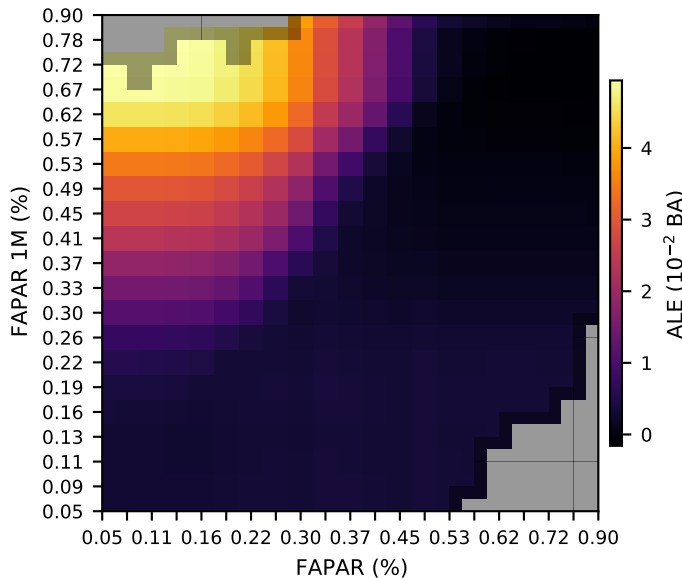

**Figure 6.** Second-order ALE plot showing the combined zeroth order (mean), first order, and second order modelled effects of FAPAR and FAPAR 1M on BA from the 15VEG_FAPAR model, taking into account all other variables. Grey boxes indicate missing data. See Fig. S13 for the sample count matrix which demonstrates the correlation between the variables and thus shows that samples are unlikely to fall into the top-left or bottom-right bins. Evenly spaced quantiles were used in the construction and labelling of the plots. It can be seen that the combined effect of FAPAR and FAPAR 1M on BA is positive if FAPAR is low while FAPAR 1M is high.

up during the preceding months is a prerequisite for fire to occur. The strong autocorrelation between current and preceding FAPAR values means that the occurrence of low current FAPAR coupled with high antecedent FAPAR is not widespread, being largely confined to shrublands in Africa. However, there is a significant interaction between current DD and current FAPAR (Fig. S14), with positive reinforcement of their mutual influence on BA when DD is high and FAPAR is low and a negative influence on BA when DD is high and FAPAR is high. Increased BA for high DD and low FAPAR is consistent with strong drought-induced fire in low productivity environments of sub-Saharan Africa, northern Australia, and isolated regions bordering the African tropical rainforests. These findings therefore support previous results, e.g. by van der Werf et al. (2008) for Australia, where it was found that antecedent precipitation coupled with instantaneous drying was important for fire activity. Decreased BA as a result of increased DD and increased FAPAR is likely a sign of high-productivity environments that are not fire-prone, despite occasional drought.

### 3.4.3 Geographically varying timescales of importance

The timescales of both fuel build-up and fuel drying are influenced by fuel type and are therefore expected to vary across biomes. Current fuel-related vegetation properties, such as FAPAR (Fig. 7a), have an important effect in tropical regions,

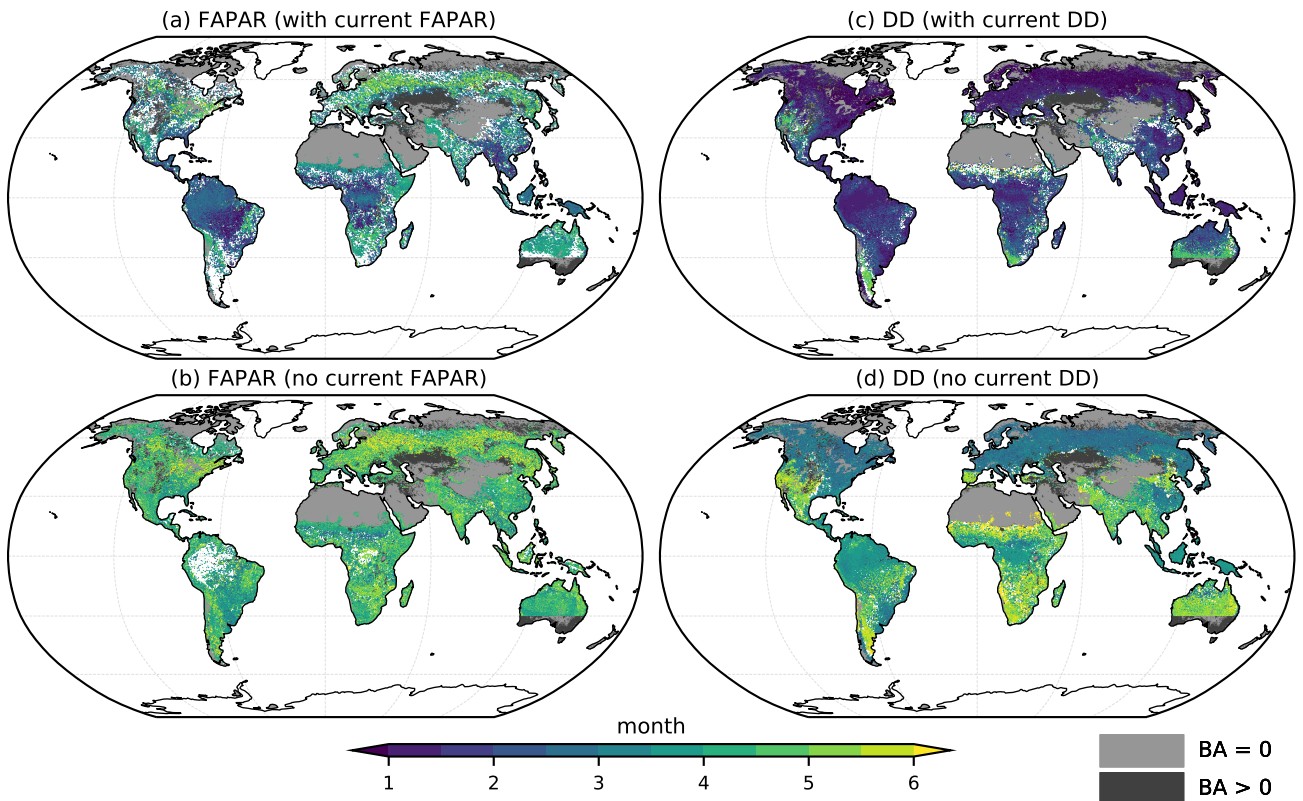

**Figure 7.** Timescales of influence of FAPAR and DD on BA. The plots show the period that is most important for determining BA from the 15VEG_FAPAR model for (a) FAPAR and (c) DD. Plots (b) and (d) show which antecedent period is most important by disregarding the influence of current conditions during plotting. Moister biomes are seen to be more influenced by current FAPAR (a), while current DD has a large influence globally (c). Note that the maps give no indication about the sign of the influence of the predictor on BA (see Fig. S10). Note also that sharp data availability boundaries (e.g. in western Asia, southern Australia) are introduced by the AGB dataset. Grey shading indicates regions with fire data availability, but where one or more of the other datasets is not available. Light grey indicates regions where mean BA is 0, with dark grey representing regions with non-zero mean BA.

particularly dry tropical regions, but are less important in temperate forest regions. Antecedent FAPAR (Fig. 7b) is important in most regions, with the strongest influence from the antecedent 3–6 months. Current DD (Fig. 7c) is generally more important than antecedent DD, although the impact on BA varies geographically: tropical and boreal regions show decreased BA as a result of low current DD, while northwestern Australia, extra-tropical Africa, the Cerrado of Brazil, and the western USA experience increased burning as a result of high current DD (Fig. S10c). The timescale on which antecedent drought affects BA (Fig. 7d) is more variable than that for fuel-related vegetation properties, ranging from 1–3 months in boreal forests, parts of sub-Saharan Africa, and northern Australia, to ∼4 months in the tropics, and ∼6 months or longer in more arid regions.


# 4    Conclusions

By using random forest algorithms to model the dependence of BA on multiple climatic and biophysical variables, we have
shown that antecedent vegetation conditions that influence fuel build-up and antecedent conditions that influence fuel drying
significantly improved model performance when predicting BA in a given month. FAPAR was shown to be the most significant
vegetation variable, and only a single vegetation variable is required for accurate BA prediction if antecedent conditions are
included. Dry-day period and maximum temperature were the most significant climatic variables influencing BA. This supports
previous studies which have shown that current climate and vegetation properties are important overall determinants of BA (e.g.
Aldersley et al., 2011; Bistinas et al., 2014; Forkel et al., 2017, 2019a; Joshi and Sukumar, 2021). The influence of antecedent
climate conditions on both fuel buildup and fuel drying has also been identified as crucial in many regions (e.g. Van Wilgen
et al., 2000; Griffin et al., 1983; Westerling et al., 2003; Swetnam and Betancourt, 1998; Jenkins et al., 2020; Archibald
et al., 2009; Krawchuk and Moritz, 2011; van der Werf et al., 2008; Randerson et al., 2005; Spessa et al., 2005). Indeed, the
geographical patterning of BA can be linked to the spatial variability of fuel loads and fuel moisture (Archibald et al., 2009;
Boer et al., 2021).

Our model-based analyses allowed us to distinguish between the immediate and antecedent impacts of fuel loads and fuel
dryness on BA, while also allowing their relative contributions to be determined. We have further shown that current and
antecedent conditions can influence BA in opposite but intuitively understandable ways: for example, wet conditions in an-
tecedent months lead to more fuel buildup in fuel limited regions and promote increased BA whereas wet conditions during any
given month reduce fuel dryness and thus limit BA. Furthermore, we have demonstrated that antecedent conditions $\geq 1$ year are
not important on a global scale. A similar conclusion was reached by Forkel et al. (2017). Important and intuitive interactions
between instantaneous and antecedent variables were captured by the models, for example supporting previous findings that
increased antecedent productivity (FAPAR) coupled with instantaneous drying (Dry Days) promotes fire activity (e.g. van der
Werf et al., 2008).

A clear contrast between fuel- and moisture-limited regions was also identified using the spatial variation of the relation-
ship between antecedent FAPAR and BA. The critical timescales involved varied with vegetation type; longer timescales
($\sim$4 months) were more important for fuel build-up in temperate regions while recent conditions were more important for
fuel drying in the tropics. The effect of vegetation variables is also biome-dependent because of differing climatic constraints.
The length of the dry-day period in the current month had the largest impact on BA but antecedent DD was also important,
particularly in temperate regions.

Future work could re-calibrate the models for each set of variables to potentially improve their performance. Approaches like
B-LOO CV or our combined variable importance metric could also be used more extensively to select the predictor variables
and potentially tune the model hyperparameters, given that these methods are currently very computationally intensive. Finally,
the findings presented herein have the potential to improve the modelling of fire at a global scale in order to improve the way
that Earth System Models depict the interactions between climate, vegetation, and fire.

*Code availability.* Computer code can be found in the empirical-fire-modelling package (Kuhn-Régnier, 2021a). ALE plots were generated using the ALEPython package (Kuhn-Régnier et al., 2021). Data analysis was carried out using the Python 3.7 (Van Rossum and Drake, 2009) packages SciPy (Virtanen et al., 2020), Matplotlib (Hunter, 2007), NumPy (Oliphant, 2006), Iris (Met Office, 2010), Dask (Dask Development Team, 2016), Jupyter notebooks (Kluyver et al., 2016), wildfires (Kuhn-Régnier, 2021b), and era5analysis (Kuhn-Régnier, 2020). GFED4 data was read using pyhdf (https://github.com/fhs/pyhdf, wraps NCSA HDF version 4).

*Author contributions.* Conceptualization, AKR, AV, ICP, SPH; Methodology, AKR; Software, AKR; Validation, AKR; Formal Analysis, AKR; Investigation, AKR, SPH; Data Curation, AKR, MF; Writing – Original Draft, AKR; Writing – Review & Editing, AKR, AV, PN, MF, ICP, SPH; Visualization, AKR; Supervision, AV, PN, ICP, SPH.

*Competing interests.* The authors declare that they have no conflict of interest.

*Acknowledgements.* AKR acknowledges support from the NERC Centre for Doctoral Training in Quantitative and Modelling skills in Ecology and Evolution (QMEE, grant reference NE/P012345/1). ICP acknowledges support from the ERC-funded project REALM (grant number 787203). SPH acknowledges support from the ERC-funded project GC2.0 (Global Change 2.0: Unlocking the past for a clearer future, grant number 694481). This research was partially funded by the Leverhulme Centre for Wildfires, Environment, and Society through the Leverhulme Trust, grant number RC-2018-023. The authors would like to thank Kees Klein Goldewijk for contributing an updated HYDE dataset.

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
