# Peer review of "The Importance of Antecedent Vegetation and Drought Conditions as Global Drivers of Burnt Area"

_Biogeosciences, 2020_

## Referee Comment (RC1) · Anonymous Referee #1 · 23 Jan 2021

Review of "Quantifying the Importance of Antecedent Fuel-Related Vegetation Properties for Burnt Area using Random Forests" by Alexander Kuhn-Régnier

The study tries to quantify the importance of antecedent vegetation status as drivers of global burnt area. The study builds on previous research using random forest modeling to understand the drivers of burnt area. While the importance of antecedent fuel load and type as drivers have been indicated previously, the study is well conducted and provides a deeper understanding towards the importance of these variables. The manuscript is well written. I have a couple of suggestions which I hope will improve the manuscript further.

[Figure]

The paper shows clearly that including indicators of fuel quantity and properties improves the representation of monthly burnt area. However, monthly burnt area includes the spatial pattern, seasonality and interannual variability all together. One thing I would have like to see was a figure trying to split these factors apart, so showing whether the inclusion of antecedent fuel indicators also improves the seasonality and/or IAV of burnt area. Now, everything is mixed together, and it is hard to know whether the results are caused by an overall improvement of the spatial pattern in burnt area, the improved representation of the seasonality or improved representation of IAV. The problems with representing seasonality and IAV of burnt area is one of the topics discussed in the introduction, and seemingly partly why the authors conducted the study, so it is a bit strange that no detailed results are presented on this topic.

I was surprised to read that the study only used data for 2010-2015, while the MODIS record now covers 20 years. Knowing that in fuel limited semi-arid regions wet events can be very sporadic, I wondered whether this short timespan does not limit the study too much, especially with regard to representing IAV. The authors indicate that they use the time period for which all variables are present, but a slightly more restrictive set of variables (e.g. the least important ones) might allow for a much longer timeseries to be used and hence present more robust results. This might also help to extract results regarding seasonality and IAV.

While I think it is nice that the authors present the training and validation model performance results (Figure 1), I cannot deny that I am a bit worried regarding the relatively large differences in model performance between the training and the validation data. One indeed always expect some difference, but (at least for the models I have been building) the differences never get so big except when there is some important overfitting going on.

Some minor comments:

Title: the title reads a bit weird, at least the "for Burnt Area" part. Maybe "as drivers of

burnt area", or something similar might sound slightly better?

L13: "are more sensitive to current conditions "are you still talking about the length of the period which needs to be considered to account for fuel build-up, or more about fuel dryness?

Table 1: "End" date seems pretty arbitrary, e.g. GFED4 burnt area has been updated up to the present.

Figure 2: There seems to be a couple of issues with the figure: 1) there is no separation between areas without data and no fire (e.g. Sahara compared to S-Australia). 2) There seems to be an artefact in the Iran/Afghanistan area, with a block-shape present. 3) for plot c the colors blend into each other so that it is hard to see any pattern (if it is present).

L215: there are a lot of abbreviations used in the manuscript. Many are pretty obvious (e.g. MaxT), but I would suggest writing out completely the less obvious ones like DD.

---

## Short Comment (SC1) · 21 Feb 2021

The manuscript "Quantifying the importance of antecedent fuel-related vegetation properties for burnt area using random forests" quantifies biophysical drivers of burned area across the globe with a particular focus on understanding how characterization of fuel build-up (and likely curing of fine fuels) in the months leading up to fire contribute to prediction in variability of area burned. The study is important and timely in that it aims to improve models of fire activity at a global scale, relevant to global fire-vegetation-climate system. The results of the study improve our knowledge of fire-fuel-climate relationships and the geography of them. In general, this is a robust study that appears

to focus mostly on the novelty of the modelling aspects rather than novelty in what is learned ecologically. Along the lines of the latter, ecological learning and context, I feel the authors could improve the introduction and discussion substantially for the modeller and non-modeller audience by delving in more depth to the wide range of existing fire studies that have asked this same question about antecedent conditions and fire activity, and placed the work and findings within that context.

Also, many of the figure captions are very hard to digest. Please consider providing the take home message to the reader to help them work through the often dense load of abbreviations and description. In other words, hold the reader by the hand. E.g., for most readers, Figure 6 caption is close to cryptic?

Line comments:

Title. I wonder if a title that describes the key findings might be more interesting than the current, methods-related title?

l. 21. The sentence on human impacts seems out of place with flow of ideas.

l. 27. This jump between fire events and fire regimes would benefit from more detail. Perhaps avoid the fire regime terminology here, and focus on events only?

l.29. not clear what is meant here by climate becoming increasingly important. It is important. Do you mean more-so that climate change will increase fire activity/severity?

ll.19-32. This paragraph has a lot of material packed in that would benefit from clearer organization and focus.

ll. 53-62. Seems like connecting the ideas to the global work on a similar topic e.g., by Krawchuk and Moritz (2011) https://doi.org/10.1890/09-1843.1, and references therein, would be helpful and warranted. I know this might seem self-centred, but it's actually pointedly relevant.

l. 105. Why would you fill those data gaps with the minimum value? I don't easily follow

the rationale. If less than 50% data, should these not be excluded from the dataset, or at least a median or mean be used?

ll. 122-125. I can see what you're trying to do with these simplifying equations, but they still need further explanation to help the reader understand the process.

Section 2.4 There are portions of this section that I don't understand. This is not your fault, it's just a bit over my head. But wanted to mention that I'll just need to trust the authors that the calculations are appropriate and correct.

ll.209/210. Interesting that the model can't capture the zeroes. I suspect this is largely based on uncertainty with the ignition-related variables?

l. 234. Interesting that there isn't a pattern.

Ll 252/253. This is very neat. And the geography is interpretable. Excellent.

ll. 299/300. Could you please make it clearer what the new learning is that we gained from this analysis. There is quite abundant literature on antecedent climate/vegetation effects on fire. You do have a novel contribution, but please highlight what it is.

l.307/308. These temporal scales of fuel build up are on the order of 50-100 years, so not really a relevant comparison to your 1-2 years timescales, is it?

l. 346. I don't follow this statement "Moisture-limited regions were more strongly affected by suppression of fire at instantaneous timescales". From what evidence is this statement based, how does this fit into your analyses and interpretation? What does it mean? Might be because this is a confusing use of the term "suppression".

---

## Short Comment (SC2) · 21 Feb 2021

My apologies, I intended my review to be posted as a RC (referee comment) rather than SC1. New to the system. Ah well, so here we are.

---

## Referee Comment (RC2) · Anonymous Referee #2 · 4 Mar 2021

The authors use a machine learning approach (ML) to investigate the impact of bio-physical and climatic variables on burned area over a five year period where there are available data on a global scale. FAPAR appears to be the most important predictor from their RF simulations. Variables were tested for their impact if they included the preceeding X number of months. This allowed for the investigation of the impact of lagged relationships, arguably very important for fire modelling.

I am concerned about the cross-validation strategy employed here (L137. By randomly choosing the validation dataset the potential for spatial autocorrelation issues arises. This is well known in the literature (see Roberts et al. 2017; Ploton et al. 2020; Kühn

_

and Dormann, 2012; Meyer et al. 2019). Here is a snippet from the abstract to the Roberts article: 'Ecological data often show temporal, spatial, hierarchical (random effects), or phylogenetic structure. Modern statistical approaches are increasingly accounting for such dependencies. However, when performing cross-validation, these structures are regularly ignored, resulting in serious underestimation of predictive error. One cause for the poor performance of uncorrected (random) cross-validation, noted often by modellers, are dependence structures in the data that persist as dependence structures in model residuals, violating the assumption of independence. Even more concerning, because often overlooked, is that structured data also provides ample opportunity for overfitting with non-causal predictors.'. Because the authors devote considerable space to discussion of these predictors, I think this issue is worth consideration. The authors also argue the the gap in R2 of the training-validation simulations gives an idea of the generalizability of the model - but that breaks down if there are spatial autocorrelation issues. Also there is some spatial structure in their biases (Fig S2) that could be coming from this issue. I would suggest adopting other CV strategies as outlined in the papers I list above.

As I am not yet convinced by their CV strategy, which is important as it impacts the results quite heavily, I suggest major revisions as I assume it will take a bit of work to demonstrate that the chosen CV strategy doesn't give misleading results.

Minor comments: - Line 78: I am not sure if I understand the DD calculation. If you had, say 5 days in one month below the precip threshold then 10 into the next. A brief precip event then the rest of the month below threshold. How does that rate? Would it be concatenated so the whole month is seen as being DD?

- L80: Soil moisture was done how? Was it percent of saturation? relative to field capacity? Or some sort of index since it is later refered to as SWI.

- L81 - The Kaplan and Lau reference is for the WGLC that is based on the WWLLN, not the WWLLN directly. It also says it provides the frequency of lightning flashes per

unit area but doesn't specify that these are ground strikes. Can you please check that these are indeed cloud to ground and not total (cloud to ground + cloud to cloud)

- Table 1: For the AGB datasets, was there any overlapping latitudes between the two datasets? If so, how was that dealt with?

- L105 - I am concerned about the gap-filling approach.So for SWI doesn't this mean that it would assume drought conditions? How often would you have this condition applied (outside of winter, L100)?

- L110: I don't follow what was done here. What do those numbers mean? This is the smallest area of herb or crop in a pixel?

- L125: I think X is referring to both the variables (FAPAR, etc.) and a single month, i.e. I think you are saying variable X at month, t, has the seasonal cycle for variable X (called in the text, X 12M) subtracted from it? Please reconsider how this is formulated in the text. This doesn't work as written.

-Table 2: To help choose which variables were useful for the ML algorithm (and which might only be contributing to overfitting), why did the authors not try applying one of the most standard techniques available such as recursive feature elimination with cross-validation (available in scikit-learn, which the authors are already using (L132); Pedregosa et al. 2011)? These techniques could help get around arbitrary choices about the number of predictors, e.g. why 15 and not 10 or 20?

- Why does Fig S2/2/3/7/etc. have straight line cutoffs? Top of Mexico, Australia, E of Iran. Plotting error? Real problem? I don't recall this being mentioned in the main text but it is in all figures. Fig 3-Also for v. low BA, what about grey instead of black? Would be easier to see the larger BA values...

- L 300- Are those the right refs? Both papers use Causality Analysis and not regression techniques as mentioned here. What aspects of those papers touches on inclusion of extra predictors and overfitting in a ML approach?

[Figure]

- p 12 has a lot of 'discussion' in the results section. Either have a 'results and discussion' section or keep them separate.

- Fig 6 - can you not use sci notation for the numbers? It makes it easier to read. I am not sure if this warrants main text inclusion as it is very simple with almost no interesting structure. I would put this in supplement and move S2 or S3 into the main.

- L310 - what about masking for areas with longer fire return intervals to see if it them pops out as more important?

Literature cited:

Roberts, D. R., Bahn, V., Ciuti, S., Boyce, M. S., Elith, J., Guillera-Arroita, G., Hauenstein, S., Lahoz-Monfort, J. J., Schröder, B., Thuiller, W., Warton, D. I., Wintle, B. A., Hartig, F. and Dormann, C. F.: Cross-validation strategies for data with temporal, spatial, hierarchical, or phylogenetic structure, Ecography , 40(8), 913–929, 2017.

Ploton, P., Mortier, F., Réjou-Méchain, M., Barbier, N., Picard, N., Rossi, V., Dormann, C., Cornu, G., Viennois, G., Bayol, N., Lyapustin, A., Gourlet-Fleury, S. and Pélissier, R.: Spatial validation reveals poor predictive performance of large-scale ecological mapping models, Nat. Commun., 11(1), 4540, 2020.

Kühn, I. and Dormann, C. F.: Less than eight (and a half) misconceptions of spatial analysis, J. Biogeogr., 39(5), 995-998, 2012.

Meyer, H., Reudenbach, C., Wöllauer, S. and Nauss, T.: Importance of spatial predictor variable selection in machine learning applications - Moving from data reproduction to spatial prediction, Ecol. Modell., 411, 108815, 2019.

Pedregosa, F., Varoquaux, G., Gramfort, A., Michel, V., Thirion, B., Grisel, O., Blondel, M., Prettenhofer, P., Weiss, R., Dubourg, V. and Vanderplas, J.: Scikit-learn: Machine Learning in Python, J. Mach. Learn. Res., 12, 2825–2830, 2011.

---

## Referee Comment (RC3) · Rene Orth (Referee) · 18 Mar 2021

Review of Kuhn-Regnier et al., bg-2020-409 "Quantifying the Importance of Antecedent Fuel-Related Vegetation Properties for Burnt Area using Random Forests"

This study investigates the role of antecedent fuel and moisture conditions for global temporal and spatial fire patterns represented by burnt area. Using a suite of random forest models with different sets of explanatory variables, the authors show that both antecedent moisture and fuel conditions are relevant for accurately modelling/predicting observed burnt areas. Thereby, the time scales extend over a few
months prior to the fire, with pronounced variations across biomes.
* * *
Recommendation: I think the paper requires moderate revisions.

This is an interesting analysis that is both relevant to the readership of Biogeosciences and a timely contribution to the ecohydrology-fire community. Legacy effects undoubtedly affect wildfire dynamics and can be a potential source of difficulties of state-of-the-art models to accurately capture fire dynamics across time scales. The machine learning approach in concert with various ecological and meteorological datasets is therefore well suited to study the underlying relationships without prior assumptions to finally provide valuable insights for the development of physically-based models. However, I have some concerns regarding the robustness of the analysis with respect to the gap filling strategy, the employed fire dataset and the relatively short analysis time period, which should to be addressed before the paper is published in Biogeosciences.
* * *
General comments:

(1) While I recognize the necessity to perform gap filling for the random forest approach in this study, I do not really like the strategy. Persistent gaps are filled using minimum values which in the case of soil water index would produce artifical droughts. While I actually do not fully understand the difference between transient and persistent gaps I agree with the authors that applying a regression-based can be suitable to fill short gaps of a few months. Nevertheless, and especially for the longer gaps extending across several consecutive months I think at least the role of the gap filling for the final conclusions needs to be tested. This could be done by additionally using an alternative gap filling strategy, or by adding random noise to the gap filled values which could be scaled by the typical inter-annual dynamics of the respective month-of-year or season of the concerned metric.

(2) It is known that there are differences between fire datasets. To illustrate the robustness of the findings of this study, it would be helpful to re-compute selected key figures with the MODIS-based ESA CCI fire dataset (Chuvieco et al. 2018).

(3) I agree with the authors that the relatively short analysis time period could have a impact on the results, particularly with respect to the long legacy effects. In this context, as lightning data which limits the available time period is not employed in all experiments with 15 predictors, they could be performed with more input data covering a longer time period.

(4) I really like that different metrics are jointly used to quantify the importance of the predictors in a robust way. It would be great if the authors could add some information in the (dis?)agreement of the results between the individual importance metrics, also to inform similar future analyses.

I do not wish to remain anonymous - Rene Orth.
* * *
Specific comments:

line 8: this should be "simulated burnt area" I guess

lines 25/26: Here you could cite O et al. (2020) and/or other previous studies on related topics.

line 66: What do you mean here with "visualization techniques"?

line 117-125: So this means you are using anomalies in the case of antecedent values for DD and the vegetation productivity proxies but absolute values in the case of the current variables? While I can understand the motivation for the removal of the seasonal cycle, I feel this is inconsistent. Why not give it all the random forest model in the ALL analysis, i.e. absolute and anomaly versions of DD and the vegetation productivity proxies at current and antecedent times, and let the model decide which of these are

most relevant? This would seem more objective to me.

line 139-141: Wouldn't it be more straightforward to use the OOB score for this, instead of dividing the dataset into training and validation parts while it is relatively short anyway?

line 142-143: Please add a comment why the random forest model is not re-calibrated for each experiment where different (numbers of) predictors are used.

lines 180-181: Why only the first 300'000?

line 228: I agree with the approach to focus on the most relevant predictors, but why did you decide on using 15 rather even fewer which could probably reduce overfitting even more while still preserving most of the model skill?

line 252: I think the FAPAR impact is strongest at high levels rather than intermediate levels.

lines 273-277: I think this is a particularly nice result which could be more highlighted in the abstract or conclusions.

line 324-326: Couldn't it be a solution to test the inclusion of antecedent BA as a predictor in the random forest model?

Figures 2 and 3: More colors are needed for the color bars to enable a finer distinction of the spatial patterns.

Figures 4 and 5: It could be informative to add uncertainty ranges to the curves, for example by re-running the random forest models many times.

Figure 5, caption: "LAI" should probably be removed in "First-order LAI ALEs"; furthermore it is not explained what is meant by first-order and second-order.

Figures 2-6: Please adapt the value labels of the axes and color bars to avoid the use of the exponent term to improve readability.

Figure 6: Why are there darker colors surrounding the gray area in the upperleft corner?

Figure 7: Why is there no data in the southern half of Australia?

References:

Chuvieco, E. et al. Generation and analysis of a new global burned area product based on MODIS 250m reflectance bands and thermal anomalies, Earth Syst. Sci. Data 10, 2015–2031 (2018).

O, S. et al. Observational evidence of wild fire-promoting soil moisture anomalies, Sci. Rep. 10, 11008 (2020).

---

## Author Comment (AC1) · 8 Apr 2021

We thank the reviewer for their constructive feedback which will greatly help to improve the quality of the paper.

Referee comments are cited in *italics* and author's responses in normal font. Responses are separated by horizontal lines.
* * *
*The study tries to quantify the importance of antecedent vegetation status as drivers*

*of global burnt area. The study builds on previous research using random forest modelling to understand the drivers of burnt area. While the importance of antecedent fuel load and type as drivers have been indicated previously, the study is well conducted and provides a deeper understanding towards the importance of these variables. The manuscript is well written. I have a couple of suggestions which I hope will improve the manuscript further.*
* * *
*The paper shows clearly that including indicators of fuel quantity and properties improves the representation of monthly burnt area. However, monthly burnt area includes the spatial pattern, seasonality and interannual variability all together. One thing I would have like to see was a figure trying to split these factors apart, so showing whether the inclusion of antecedent fuel indicators also improves the seasonality and/or IAV of burnt area. Now, everything is mixed together, and it is hard to know whether the results are caused by an overall improvement of the spatial pattern in burnt area, the improved representation of the seasonality or improved representation of IAV. The problems with representing seasonality and IAV of burnt area is one of the topics discussed in the introduction, and seemingly partly why the authors conducted the study, so it is a bit strange that no detailed results are presented on this topic.*

*I was surprised to read that the study only used data for 2010-2015, while the MODIS record now covers 20 years. Knowing that in fuel limited semi-arid regions wet events can be very sporadic, I wondered whether this short timespan does not limit the study too much, especially with regard to representing IAV. The authors indicate that they use the time period for which all variables are present, but a slightly more restrictive set of variables (e.g. the least important ones) might allow for a much longer timeseries to be used and hence present more robust results. This might also help to extract results regarding seasonality and IAV.*

Mentioning the IAV in the introduction is a motivation for our study, but we acknowledge

that our study did not allow us to look at it previously.

In our revised manuscript, we make an attempt to disentangle the improvement to the representation of IAV using an analysis of monthly data (as opposed to climatological averages) for the time period 11-2000ăĬ12-2019 (230 months), which also expands upon the limited temporal range as requested by the reviewer. A different burnt area dataset, MODIS MCD64CMQ BA, was also used for this run. This experiment, the 15VEG_FAPAR_MON experiment, otherwise uses the same variables as the 15VEG_FAPAR experiment.

These results show decreased R2 values, which is to be expected given the higher variance of this data:

- Using random cross-validation; Test R2: 0.45, OOB R2: 0.45

- Excluding the years 2009-2012 (including 2012); Test R2: 0.38, OOB R2: 0.45

- Excluding the years 2016-2019 (including 2019); Test R2: 0.41, OOB R2: 0.45

For this temporal cross-validation either period 2009-2012 or period 2016-2019 is left out for testing.

The main relationships identified by the model did not change appreciably, however, showing that there is no temporal change that is important. The spatial patterns are dominant, since the models behave very similarly when comparing climatological and monthly data, and the main commonality between those data is the geographical pattern.

For example, the FAPAR ALE plots for the 15VEG_FAPAR (climatological) and 15VEG_FAPAR_MON (monthly, expanded temporal range) models are shown in Figures 1 and 2, respectively.

These ALEs are highly consistent. Note that the smaller variability in the second plot (monthly analysis) arises from the fact that a constant proportion of bootstrap samples is used to construct the shaded region, and since the monthly data has more samples, these random samples are larger.

Other variables are similarly consistent, e.g. DD 3M as shown in Figures 3 and 4.
* * *
*While I think it is nice that the authors present the training and validation model performance results (Figure 1), I cannot deny that I am a bit worried regarding the relatively large differences in model performance between the training and the validation data. One indeed always expect some difference, but (at least for the models I have been building) the differences never get so big except when there is some important overfitting going on.*

We recognise that looking just at the training (in-the-bag) and validation scores is not valid, which is why we also include the OOB scores in our revised manuscript. These provide a good indication of how the model is expected to generalise to unseen data (Fox et al., 2017). Considering the construction of the random forest models – a set of decision trees which, in principle, can fit all training data perfectly – it is expected that some overfitting will occur even after cross-validation and given that the algorithm mitigates overfitting by averaging over the predictions of many decision trees (i.e. the forest). Since our OOB scores and the validation scores are similar (see Fig. 5), this indicates that the model is not overfitting.

Furthermore, there is no agreed standard for considering whether a model is robust or not. However, values of explained variance above 60% (i.e. an $R^2$ score of 0.6) have been used to indicate this (e.g. Thomas et al., 2014). Both the OOB and validation set $R^2$ scores are above this threshold for the relevant models.

As an additional assurance of model robustness, we present the PFI importances computed separately on the training and test sets in Fig. 6. The fact that there is no appreciable difference between the two is an indicator of the lack of overfitting by the model, since the model has not learnt to consider certain variables more important than they actually are during training (Dankers and Pfisterer, 2020).
* * *
*Some minor comments:*
* * *
*Title: the title reads a bit weird, at least the "for Burnt Area" part. Maybe "as drivers of burnt area", or something similar might sound slightly better?*

The title has been changed to "The Importance of Antecedent Vegetation and Drought Conditions as Global Drivers of Burnt Area" to reflect the key findings more closely.
* * *
*L13: "are more sensitive to current conditions" are you still talking about the length of the period which needs to be considered to account for fuel build-up, or more about fuel dryness?*

We have updated the abstract to clarify that the current conditions refer to fuel dryness instead of fuel build-up.
* * *
*Table 1: "End" date seems pretty arbitrary, e.g. GFED4 burnt area has been updated up to the present.*

As far as we are aware, the GFED4 dataset without small fires has not been updated since we conducted our original study. Therefore, we have used the MODIS MCD64CMQ burnt area dataset to carry out modelling studies up to and including 12-2019. We have updated other variables in the table to reflect their evolving nature (e.g.

MODIS data, which is continually being updated), instead of only reflecting the state of our processed data.
* * *
*Figure 2: There seems to be a couple of issues with the figure: 1) there is no separation between areas without data and no fire (e.g. Sahara compared to S-Australia). 2) There seems to be an artefact in the Iran/Afghanistan area, with a block-shape present. 3) for plot c the colors blend into each other so that it is hard to see any pattern (if it is present).*

As shown in Fig. 7, we have added grey shading to indicate regions with fire data availability, but where one or more of the other datasets is not available. Light grey indicates regions where mean BA is 0, with dark grey representing regions with nonzero mean BA. The 'block-shapes' originate from the AGB dataset used, which we have indicated in all relevant captions in the updated manuscript. Note that in addition to the shown changes in this figure, the colour scheme for plot c will also be revised to make the differences more apparent, and the number of divisions adjusted in order to make the plots clearer.
* * *
*L215: there are a lot of abbreviations used in the manuscript. Many are pretty obvious (e.g. MaxT), but I would suggest writing out completely the less obvious ones like DD.*

We include many abbreviations, but since it is difficult to distinguish common from uncommon abbreviations, we previously included a table with definitions (Table 1). We have now improved the presentation of abbreviations in Table 1 by listing each variable along with its abbreviation on a separate line. This should hopefully make it easier to discern which abbreviation corresponds to which variable and dataset.

**References**

Dankers, Cord, and Florian Pfisterer. 2020. Chapter 11 PFI: Training vs. Test Data | Limitations of Interpretable Machine Learning Methods. https://compstat-lmu.github.io/iml_methods_limitations/pfi-data.html#pfi-data.

Fox, Eric W., Ryan A. Hill, Scott G. Leibowitz, Anthony R. Olsen, Darren J. Thornbrugh, and Marc H. Weber. 2017. 'Assessing the Accuracy and Stability of Variable Selection Methods for Random Forest Modeling in Ecology'. Environmental Monitoring and Assessment 189 (7): 316. https://doi.org/10.1007/s10661-017-6025-0.

Thomas, P. B., P. J. Watson, R. A. Bradstock, T. D. Penman, and O. F. Price. 2014. 'Modelling Surface Fine Fuel Dynamics across Climate Gradients in Eucalypt Forests of South-Eastern Australia'. Ecography 37 (9): 827–37. https://doi.org/10.1111/ecog.00445.

―――――――――――――――――――――――

[Figure]

**Fig. 1.** First-order ALE plot showing the effect of instantaneous FAPAR on burnt area (BA) in the 15VEG_FAPAR model after accounting for all other variables.

[Figure]

**Fig. 2.** First-order ALE plot showing the effect of instantaneous FAPAR on burnt area (BA) in the 15VEG_FAPAR_MON model after accounting for all other variables.

[Figure]

**Fig. 3.** First-order ALE plot showing the effect of the 3-month antecedent dry-day period (DD 3M) on burnt area (BA) in the 15VEG_FAPAR model after accounting for all other variables.

[Figure]

**Fig. 4.** First-order ALE plot showing the effect of the 3-month antecedent dry-day period (DD 3M) on burnt area (BA) in the 15VEG_FAPAR_MON model after accounting for all other variables.

[Figure]

**Fig. 5.** Global climatological R2 scores for the different experiments. Note that although train R2 scores are shown here, train OOB scores are more indicative of model performance on unseen data.

[Figure]

**Fig. 6.** PFI importances for the 15VEG_FAPAR model for the training and test sets. Note that the error bars originate from repeated shuffling of the investigated variable as opposed to different datasets.

[Figure]

**Fig. 7.** (a) Average observed (Ob.) BA derived from the GFED4 BA dataset. (b) Out-of-sample predictions (Pr.) by the ALL model. (c) Relative prediction error of the ALL model.

---

## Author Comment (AC2) · 8 Apr 2021

Dear Meg Krawchuk,

Thank you for your positive and constructive review, which we feel will greatly improve the paper. Below we have outlined the changes we will make to address the points raised.

Referee comments are cited in *italics* and author's responses in normal font. Responses are separated by horizontal lines.

*The manuscript "Quantifying the importance of antecedent fuel-related vegetation properties for burnt area using random forests" quantifies biophysical drivers of burned area across the globe with a particular focus on understanding how characterization of fuel build-up (and likely curing of fine fuels) in the months leading up to fire contribute to prediction in variability of area burned. The study is important and timely in that it aims to improve models of fire activity at a global scale, relevant to global fire-vegetation-climate system. The results of the study improve our knowledge of fire-fuel-climate relationships and the geography of them. In general, this is a robust study that appears to focus mostly on the novelty of the modelling aspects rather than novelty in what is learned ecologically. Along the lines of the latter, ecological learning and context, I feel the authors could improve the introduction and discussion substantially for the modeller and non-modeller audience by delving in more depth to the wide range of existing fire studies that have asked this same question about antecedent conditions and fire activity, and placed the work and findings within that context.*

*Also, many of the figure captions are very hard to digest. Please consider providing the take home message to the reader to help them work through the often dense load of abbreviations and description. In other words, hold the reader by the hand. E.g., for most readers, Figure 6 caption is close to cryptic?*

In the updated manuscript, previous studies investigating antecedent conditions will be mentioned more in the introduction and discussion (see further responses below).

Regarding figure captions, additional information has been added in order to point out the most significant features more clearly. Furthermore, the introduction of ALE plots has been modified to remove the discrepancy between e.g. "1D" and "first-order" ALE plots, which are both referring to the same thing, thereby introducing the differences between first- and second-order plots in the Methods section. For example, we have updated the caption for Fig. 6 (in the manuscript): "Second-order ALE plot showing the

combined zeroth order (mean), first order, and second order modelled effects of FAPAR and FAPAR 1M on BA from the 15VEG_FAPAR model, taking into account all other variables. Grey boxes indicate missing data. See Fig. S7 for the sample count matrix which demonstrates the correlation between the variables and thus shows that samples are unlikely to fall into the top-left or bottom-right bins. Evenly spaced quantiles are used in the construction and labelling of the plots. It can be seen that the combined effect of FAPAR and FAPAR 1M on BA is positive if FAPAR is low while FAPAR 1M is high."

*Line comments:*

*Title. I wonder if a title that describes the key findings might be more interesting than the current, methods-related title?*

The title has been changed to "The Importance of Antecedent Vegetation and Drought Conditions as Global Drivers of Burnt Area" to reflect the key findings more closely.

*l. 21. The sentence on human impacts seems out of place with flow of ideas.*

This sentence was originally included to highlight the multiplicity of factors involved in controlling wildfires. This paragraph has been rewritten to further pronounce the climatic and ecological factors that are the dominant topics of our discussion, while mentioning these differential human impacts below.

*l. 27. This jump between fire events and fire regimes would benefit from more detail. Perhaps avoid the fire regime terminology here, and focus on events only?*

In this instance, we will replace "fire regimes" → "fire activity" to avoid confusion. The following sentence refers to a decrease in global mean burnt area and therefore should clarify that we are interested in large-scale patterns.
* * *
*l.29. not clear what is meant here by climate becoming increasingly important. It is important. Do you mean more-so that climate change will increase fire activity/severity?*

Considering the relative importance of several wildfire drivers (e.g. climate, vegetation structure, or ignitions), we meant to indicate that the relative importance of climatic influence on wildfires is predicted to increase in certain regions. We have now removed this sentence to increase the clarity of the introduction and focus more on wildfires drivers and their interactions.
* * *
*ll.19-32. This paragraph has a lot of material packed in that would benefit from clearer organization and focus.*

This paragraph has been reorganised. The sentence about human impacts on line 21 has been moved to a new paragraph, which discusses the impact of climate change on fire. Thus, the biophysical drivers of fire (and their coupling) are discussed separately.
* * *
*ll. 53-62. Seems like connecting the ideas to the global work on a similar topic e.g., by Krawchuk and Moritz (2011) https://doi.org/10.1890/09-1843.1, and references therein, would be helpful and warranted. I know this might seem self-centred, but it's actually pointedly relevant.*

We have added this paragraph about previous studies to the introduction:

"A number of regional and global studies have indicated the importance of antecedent fuel build-up for BA. For example, links between fire activity and antecedent productivity

have been found in South Africa (Van Wilgen et al., 2000), central Australia (Griffin et al., 1983), grass and shrublands in the western US (Littell et al., 2009; Westerling et al., 2003; Swetnam and Betancourt, 1998), NSW Australia (for bushfire fuel) (Jenkins et al., 2020), and southern Africa (Archibald et al., 2009). Global studies have identified similar relationships (a positive relationship between pre-season productivity and fire activity in the following dry season) in some dry areas. By studying the correlation between growing period (i.e. antecedent) soil moisture and fire activity, Krawchuk and Moritz (2011) found fire activity in dry regions to be related to antecedent productivity. Similarly, van der Werf et al. (2008) found a similar relationship for arid ecosystem (e.g. N. AUS), where antecedent wet conditions coupled with instantaneous drying were important. Other global studies have identified northern Australia as obeying this relationship too (Randerson et al., 2005; Spessa et al., 2005). In a more recent global analysis, O et al. (2020) found that for arid regions, wet anomalies (soil moisture) lead to increased fire later in the year by increasing fuel loads and biomass. Thus, it is clear that a better understanding of the timescales of fuel accumulation, the interaction between biophysical drivers and fuel build-up, and the effects of antecedent weather conditions on both fuel loads and fuel drying is needed in order to improve predictions of BA."
* * *
*l. 105. Why would you fill those data gaps with the minimum value? I don't easily follow the rationale. If less than 50% data, should these not be excluded from the dataset, or at least a median or mean be used?*

The algorithm used to discern the location of 'permanent' as opposed to 'transient' gaps utilises the amount of missing information for a specific month at each location. For example, if a certain grid cell was missing data for more than 50% of all Decembers in the record, these gaps in December would be treated as a permanent gap and therefore subject to filling by minimum values. Remaining gaps are treated as transient and therefore filled using the regression model outlined in the text.

Unfortunately, simple exclusion of the times lacking data is not possible for our analysis because we rely on antecedent samples throughout. Thus, data gaps (which predominantly affect extreme latitudes in winter due to snow cover, but also occur due to cloud cover or limitations of passive satellite sensors in winter at extreme latitudes) have to be filled in order to allow analysis of summer months at those locations.

The algorithm we use to fill data gaps for the SWI, FAPAR, LAI, SIF, and VOD datasets is based on an algorithm proposed by Forkel et al. (2017). If 50% or more of the observations at a given location and month are missing, we assume that this indicates missing data during the winter, since other causes for data gaps (e.g. cloud cover) are assumed to be predominantly transient. Note also that locations with very little data (regardless of whether such data gaps always occur in the same month, as above, or at any point throughout the year) are discarded if no observations are available for more than 52 months out of the total 88 months investigated.

The winter data gaps identified above are then filled using the minimum value observed at that location because of the seasonality of the filled variables. For example, FAPAR would be expected to be at its minimum during the winter. Consequently, we use the minimum observed value to indicate a low FAPAR value during this time.

As can be seen in Fig. 1, virtually no samples are being filled with the minimum value outside of winter, as we assume above.
* * *
*ll. 122-125. I can see what you're trying to do with these simplifying equations, but they still need further explanation to help the reader understand the process.*

We have added the sentence "For example, the 12-month antecedent X 12M was transformed by subtracting the instantaneous (month 0) value of X, thereby yielding the anomaly in X, X $\Delta$12M, that may be easier to interpret." Additionally, we have adjusted the notation to make the distinction between the instantaneous variable (X

0M) and the variable, X, itself clearer.
* * *
*Section 2.4 There are portions of this section that I don't understand. This is not your fault, it's just a bit over my head. But wanted to mention that I'll just need to trust the authors that the calculations are appropriate and correct.*

We have rephrased parts of this section to convey the calculations more clearly.
* * *
*ll. 209/210. Interesting that the model can't capture the zeroes. I suspect this is largely based on uncertainty with the ignition-related variables?*

One of the reasons the model may struggle to predict the exact value of '0' may be because the model (random forest) consists of many smaller models (decision trees), where the final model prediction is computed as the mean of the individual predictions of the smaller models. Thus, all 500 individual models would have to predict '0' to yield this value overall, which does not appear to occur in our model, given the stochastic nature of the training process.
* * *
*l. 234. Interesting that there isn't a pattern.*

We agree that this figure is not very easy to interpret, but what it is showing is that where there are improvements, the improvements are relatively large. But there is no coherent spatial patterning in it. Nor do we necessarily expect to see a spatial patterning in the improvement, because we anticipate antecedent conditions to be relevant in most areas.

We find that over half the grid cells improve ( 56%, as stated in the manuscript), and where there are improvements, these tend to be larger than the instances of worsening performance. This can more clearly be seen in Fig. 2, where the error comparison is visualised as a histogram, with the error multiplied by –1 (i.e. <|Err(ALL)| - |Err(CURR)|>) being shown in orange to more clearly visualise the skewness of the distribution. The skewed distribution of the errors clearly demonstrates that improvements are larger, thereby leading to the overall global improvement of R2 score.
* * *
*LI 252/253. This is very neat. And the geography is interpretable. Excellent.*

We have added some additional links between these results and the SHAP value maps (Fig. 7 in the manuscript) to clarify the link to the geography.
* * *
*ll. 299/300. Could you please make it clearer what the new learning is that we gained from this analysis. There is quite abundant literature on antecedent climate/vegetation effects on fire. You do have a novel contribution, but please highlight what it is.*

We have rewritten this to highlight the novel contributions more clearly:Âă

"We have shown that antecedent vegetation conditions that influence fuel build-up and antecedent conditions that influence fuel drying strongly influence BA in a given month. Many previous studies have shown that current climate and vegetation properties are important overall determinants of BA (e.g. Aldersley et al., 2011; Bistinas et al., 2014; Forkel et al., 2017, 2019a; Joshi and Sukumar, 2021). The influence of antecedent climate conditions on fuel buildup and on fuel drying has also been identified as crucial in many regions (e.g. Van Wilgen et al., 2000; Griffin et al., 1983; Westerling et al., 2003; Swetnam and Betancourt, 1998; Jenkins et al., 2020; Archibald et al., 2009; Krawchuk and Moritz, 2011; van der Werf et al., 2008; Randerson et al., 2005; Spessa et al., 2005). Indeed, spatial variability in fuel loads and fuel moisture are important determinants of the geographical patterning of BA (Archibald et al., 2009; Boer et al., 2021). Our model-based analyses allow us to distinguish between the immediate and antecedent impacts of fuel loads and fuel dryness on BA, while also allowing their relative contributions to be determined. We have further shown that current and antecedent conditions can influence BA in opposite but intuitively understandable ways: wet conditions in antecedent months, for example, lead to more fuel buildup in fuel limited regions and promote increased BA whereas wet conditions during any given month reduce fuel dryness and thus limit BA. Furthermore, we have demonstrated that antecedent conditions >1 year are not important on a global scale. A similar conclusion was reached by Forkel et al. (2017). Furthermore, the critical timescale for fuel build-up varies with vegetation type, with longer timescales being more important in temperate regions and recent conditions being more important in the tropics. The effect of vegetation variables is also biome-dependent because of differing climatic constraints. The length of the dry-day period in the current month has the largest impact on BA but antecedent DD can also be important, particularly in temperate regions. FAPAR was shown to be the most significant vegetation variable, and only a single vegetation variable is required for accurate BA prediction if antecedent conditions are included. There are also significant, mostly intuitive, interactions between variables. For example, antecedent productivity (FAPAR) coupled with instantaneous drying (Dry Days) was determined to be important, in accordance with previous studies (e.g. van der Werf et al., 2008)."
* * *
*l.307/308. These temporal scales of fuel build up are on the order of 50-100 years, so not really a relevant comparison to your 1-2 years timescales, is it?*

We have made the impact of limiting our antecedent variables to at most 2 years clearer in the text, since this does of course prevent us from accounting for long-term fuel build-up as would be relevant for these biomes:

"The failure to detect an influence of longer-term fuel build-up on BA probably reflects the short time interval (1–2 yrs) considered for antecedent fuel build-up, far shorter than the timescales of coarse fuel build-up in these ecosystems. The seasonal differences captured by our analyses may be unimportant in regions where long fire return times

(or fire suppression) allow fuel build-up over longer periods."
* * *
*I. 346. I don't follow this statement "Moisture-limited regions were more strongly affected by suppression of fire at instantaneous timescales". From what evidence is this statement based, how does this fit into your analyses and interpretation? What does it mean? Might be because this is a confusing use of the term "suppression".*

We have rewritten this part to avoid using "suppression" in favour of "limitation", which more accurately describes our intent; instantaneous conditions reduce fuel available to burn due to moisture. In these regions, antecedent conditions are also expected to be less important due to lack of seasonal fuel build-up patterns. This statement is based on Fig. 7a in the manuscript, where it can be seen that in moisture-limited regions, the instantaneous timescales (drying in and up to the current month) are most important for enabling fire.
* * *
**References**

Aldersley, Andrew, Steven J. Murray, and Sarah E. Cornell. 2011. 'Global and Regional Analysis of Climate and Human Drivers of Wildfire'. Science of The Total Environment 409 (18): 3472–81. https://doi.org/10.1016/j.scitotenv.2011.05.032.

Archibald, Sally, David P. Roy, Brian W. van Wilgen, and Robert J. Scholes. 2009. 'What Limits Fire? An Examination of Drivers of Burnt Area in Southern Africa'. Global Change Biology 15 (3): 613–30. https://doi.org/10.1111/j.1365-2486.2008.01754.x.

Bistinas, I., S. P. Harrison, I. C. Prentice, and J. M. C. Pereira. 2014. 'Causal Relationships versus Emergent Patterns in the Global Controls of Fire Frequency'. Biogeosciences 11 (18): 5087–5101. https://doi.org/10.5194/bg-11-5087-2014.

Boer, Matthias M., Victor Resco De Dios, Elisa Stefaniak, and Ross A. Bradstock. 2021. 'A Hydroclimatic Model for the Distribution of Fire on Earth'. Environmental Research Communications. https://doi.org/10.1088/2515-7620/abec1f.

Forkel, Matthias, Niels Andela, Sandy P. Harrison, Gitta Lasslop, Margreet van Marle, Emilio Chuvieco, Wouter Dorigo, et al. 2019. 'Emergent Relationships with Respect to Burned Area in Global Satellite Observations and Fire-Enabled Vegetation Models'. Biogeosciences 16 (1): 57–76. https://doi.org/10.5194/bg-16-57-2019.

Forkel, Matthias, Wouter Dorigo, Gitta Lasslop, Irene Teubner, Emilio Chuvieco, and Kirsten Thonicke. 2017. 'A Data-Driven Approach to Identify Controls on Global Fire Activity from Satellite and Climate Observations (SOFIA V1)'. Geoscientific Model Development 10 (12): 4443–76. https://doi.org/10.5194/gmd-10-4443-2017.

Griffin, GF, NF Price, and HF Portlock. 1983. 'Wildfires in the Central Australian Rangelands, 1970-1980.' Journal of Environmental Management 17 (4): 311–23.

Jenkins, Meaghan E., Michael Bedward, Owen Price, and Ross A. Bradstock. 2020. 'Modelling Bushfire Fuel Hazard Using Biophysical Parameters'. Forests 11 (9): 925. https://doi.org/10.3390/f11090925.

Joshi, Jaideep, and Raman Sukumar. 2021. 'Improving Prediction and Assessment of Global Fires Using Multilayer Neural Networks'. Scientific Reports 11 (1): 3295. https://doi.org/10.1038/s41598-021-81233-4.

Krawchuk, Meg A., and Max A. Moritz. 2011. 'Constraints on Global Fire Activity Vary across a Resource Gradient'. Ecology 92 (1): 121–32. https://doi.org/10.1890/09-1843.1.

Littell, Jeremy S., Donald McKenzie, David L. Peterson, and Anthony L. Westerling. 2009. 'Climate and Wildfire Area Burned in Western U.S. Ecoprovinces, 1916–2003'. Ecological Applications 19 (4): 1003–21. https://doi.org/10.1890/07-1183.1.

O, Sungmin, Xinyuan Hou, and Rene Orth. 2020. 'Observational Evidence of Wildfire-Promoting Soil Moisture Anomalies'. Scientific Reports 10 (1): 11008. https://doi.org/10.1038/s41598-020-67530-4.

Randerson, J. T., G. R. van der Werf, G. J. Collatz, L. Giglio, C. J. Still, P. Kasibhatla, J. B. Miller, J. W. C. White, R. S. DeFries, and E. S. Kasischke. 2005. 'Fire Emissions from C3 and C4 Vegetation and Their Influence on Interannual Variability of Atmospheric CO2 and $\Delta$13CO2'. Global Biogeochemical Cycles 19 (2). https://doi.org/10.1029/2004GB002366.

Spessa, Allan, Bevan McBeth, and Colin Prentice. 2005. 'Relationships among Fire Frequency, Rainfall and Vegetation Patterns in the Wet–Dry Tropics of Northern Australia: An Analysis Based on NOAA-AVHRR Data'. Global Ecology and Biogeography 14 (5): 439–54. https://doi.org/10.1111/j.1466-822x.2005.00174.x.

Swetnam, Thomas W., and Julio L. Betancourt. 1998. 'Mesoscale Disturbance and Ecological Response to Decadal Climatic Variability in the American Southwest'. Journal of Climate 11 (12): 3128–47. https://doi.org/10.1175/1520-0442(1998)011<3128:MDAERT>2.0.CO;2.

Van Wilgen, B. W., H. Biggs, S. P. O'Regan, and N. Mare. 2000. 'Fire History of the Savanna Ecosystems in the Kruger National Park, South Africa, between 1941 and 1996'. South African Journal of Science 96 (April). https://researchspace.csir.co.za/dspace/handle/10204/1890.

Werf, Guido R. van der, James T. Randerson, Louis Giglio, Nadine Gobron, and A. J. Dolman. 2008. 'Climate Controls on the Variability of Fires in the Tropics and Subtropics'. Global Biogeochemical Cycles 22 (3). https://doi.org/10.1029/2007GB003122.

Westerling, A. L., A. Gershunov, T. J. Brown, D. R. Cayan, and M. D. Dettinger. 2003. 'Climate and Wildfire in the Western United States'. Bulletin of the American Meteorological Society 84 (5): 595–604. https://doi.org/10.1175/BAMS-84-5-595.

[Figure]

**Fig. 1.** The proportion of filled samples for FAPAR, with yellow indicating that all occurrences of a given month at a given location were filled and purple indicating no filling was done.

[Figure]

**Fig. 2.** The change in burnt area (BA) prediction error magnitude between the ALL and CURR models. The change multiplied by -1 is shown in orange to more clearly visualise the skewness of the distribution.

---

## Author Comment (AC3) · 8 Apr 2021

We thank the reviewer for their detailed reading of the text and correspondingly constructive feedback which will undoubtedly improve the quality of this work.

Referee comments are cited in *italics* and author's responses in normal font. Responses are separated by horizontal lines.
* * *
*The authors use a machine learning approach (ML) to investigate the impact of bio-*

*physical and climatic variables on burned area over a five year period where there are available data on a global scale. FAPAR appears to be the most important predictor from their RF simulations. Variables were tested for their impact if they included the preceeding X number of months. This allowed for the investigation of the impact of lagged relationships, arguably very important for fire modelling.*

*I am concerned about the cross-validation strategy employed here (L137. By randomly choosing the validation dataset the potential for spatial autocorrelation issues arises. This is well known in the literature (see Roberts et al. 2017; Ploton et al. 2020; Kühn and Dormann, 2012; Meyer et al. 2019). Here is a snippet from the abstract to the Roberts article: 'Ecological data often show temporal, spatial, hierarchical (random effects), or phylogenetic structure. Modern statistical approaches are increasingly accounting for such dependencies. However, when performing cross-validation, these structures are regularly ignored, resulting in serious underestimation of predictive error. One cause for the poor performance of uncorrected (random) cross-validation, noted often by modellers, are dependence structures in the data that persist as dependence structures in model residuals, violating the assumption of independence. Even more concerning, because often overlooked, is that structured data also provides ample opportunity for overfitting with non-causal predictors.'. Because the authors devote considerable space to discussion of these predictors, I think this issue is worth consideration. The authors also argue [that] the gap in R2 of the training-validation simulations gives an idea of the generalizability of the model - but that breaks down if there are spatial autocorrelation issues. Also there is some spatial structure in their biases (Fig S2) that could be coming from this issue. I would suggest adopting other CV strategies as outlined in the papers I list above.*

Comparing Fig. S2 (in the supplementary materials) to the normalised (by mean observations) differences (Fig. 2c in the manuscript), the spatial pattern changes if not disappears. The main spatial patterns apparent from Fig. S2 (in the supplementary materials) follow the magnitude of mean BA.

We will also adjust the number of bins in Fig. 2c (in the manuscript) in order to make the plot easier to read.

Further, the question concerning robustness verification is an important one, and we thank the reviewer for mentioning it. This will be addressed further below.

―――――――――――――――――――――――――――――――――

*As I am not yet convinced by their CV strategy, which is important as it impacts the results quite heavily, I suggest major revisions as I assume it will take a bit of work to demonstrate that the chosen CV strategy doesn't give misleading results.*

We have undertaken additional experiments to demonstrate the robustness of our model against overfitting. Excluding a variable number of grid cells around the testing data grid points (as in Ploton et al. (2020)) while keeping the number of training grid points constant yields the results shown in Fig. 1.

It can be seen that the performance of the model drops as a larger region around the test samples is excluded (with 40 pixels corresponding roughly to 1200 km at the equator). However, as opposed to the case study in Ploton et al. (2020), the performance remains relatively stable, not dropping below an $R2$ score of 0.4.

We have also undertaken an analysis of monthly data (as opposed to climatological averages) for the time period 11-2000—12-2019 (230 months). A different burnt area dataset, MODIS MCD64CMQ BA, was used for this run. This experiment, the 15VEG_FAPAR_MON experiment, otherwise uses the same variables as the 15VEG_FAPAR experiment.

These results again show decreased $R2$ values, which is to be expected given the higher variance of this data:

- Using random cross-validation; Test R2: 0.45, OOB R2: 0.45

- Excluding the years 2009-2012 (including 2012); Test R2: 0.38, OOB R2: 0.45

- Excluding the years 2016-2019 (including 2019); Test R2: 0.41, OOB R2: 0.45

Despite the slightly reduced R2 values, the above experiments show that the model is able to robustly predict BA under spatial and temporal cross-validation scenarios. For the temporal cross-validation either period 2009-2012 or period 2016-2019 is left out for testing.
* * *
*Minor comments:*
* * *
*- Line 78: I am not sure if I understand the DD calculation. If you had, say 5 days in one month below the precip threshold then 10 into the next. A brief precip event then the rest of the month below threshold. How does that rate? Would it be concatenated so the whole month is seen as being DD?*

Concatenation is only carried out across month boundaries if the resulting dry-day period is contiguous. Wetting precipitation events always disrupt a dry-day period. We have added an example calculation to the manuscript to make this clearer:

"The dry-day period was defined as the longest contiguous period of ERA5 mean daily precipitation below 0.1 mm day$^{-1}$ (wetting rainfall; Harris et al., 2014; Jolly et al., 2015) within each month. A period contiguous with the previous month's dry-day period was concatenated such that the sum of both (number of days) was used to determine the longest period. For example, consider a 30-day long month with a 10-day long dry-day period at the beginning of the month, followed by a wetting precipitation event on day 11, and then a dry-day period for the following 19 days. This month has a dry-day period of 19 days. However, if the previous month were to terminate in a 10-day long dry-day period, these 10 days would be added to the initial 10-day dry-day period of the current month, thereby making this combined dry-day period the longest."

*- L80: Soil moisture was done how? Was it percent of saturation? relative to field capacity? Or some sort of index since it is later refered to as SWI.*

It is the SWI index as provided by Copernicus, see mention in line 81 of the manuscript.

*- L81 - The Kaplan and Lau reference is for the WGLC that is based on the WWLLN, not the WWLLN directly. It also says it provides the frequency of lightning flashes per unit area but doesn't specify that these are ground strikes. Can you please check that these are indeed cloud to ground and not total (cloud to ground + cloud to cloud)*

That is true, WWLLN was used previously because it is the dataset that the WGLC is based on. The manuscript has been updated to use the WGLC dataset name. We have also added additional references regarding the type of lightning strokes detected by the network; the WWLLN network is unable to rule out detection of any cloud to cloud strikes but is most sensitive to cloud to ground strikes:

"We used the WGLC dataset (Kaplan and Lau, 2019) which provides counts of monthly lightning strikes. It is based on the World Wide Lightning Location Network (WWLLN) dataset, which mainly detects cloud-to-ground strikes (Rodger et al., 2004; Abarca et al., 2010), as opposed to LIS (Bürgesser, 2017)."

*- Table 1: For the AGB datasets, was there any overlapping latitudes between the two datasets? If so, how was that dealt with?*

Both datasets were first averaged from the original resolution to a 0.25° x 0.25° grid, then mosaicked to the global extent by taking the mean. This has been pointed out in the updated manuscript, along with the original resolutions of the individual datasets:

"Tree AGB was obtained by mosaicking AGB datasets for the tropics (Avitabile et al.,

2016, 1 km resolution) and for northern forests (Thurner et al., 2014, 0.01° resolution) using the mean after resampling each to a common spatial resolution of 0.25°."
* * *
*- L105 - I am concerned about the gap-filling approach.So for SWI doesn't this mean that it would assume drought conditions? How often would you have this condition applied (outside of winter, L100)?*

As shown in Fig. 2 for FAPAR, outside of winter the minimum value filling is virtually never applied. Additionally, it is mostly limited to northern latitudes in winter, as expected.

Regarding the filling of SWI, this should not have a big influence on the final results because we are not using antecedent values of SWI in any case. Since we don't expect fires during the winter, having (by necessity of gap filling) unphysical values of SWI in the winter should not affect results where relevant for our analysis. Other variables (e.g. low temperature during the winter) should be sufficient for the random forest model to ignore these SWI values.
* * *
*- L110: I don't follow what was done here. What do those numbers mean? This is the smallest area of herb or crop in a pixel?*

They are indeed the smallest observed area of HERB or CROP in a pixel. Missing land cover was previously filled using these values to prevent excessive data gaps.

However, this has now become obsolete due to renewed processing of the landcover dataset (to enable processing of a longer time period for specific runs). Consequently, this line has been removed from the manuscript.
* * *
*- L125: I think X is referring to both the variables (FAPAR, etc.) and a single month,*

*i.e. I think you are saying variable X at month, t, has the seasonal cycle for variable X (called in the text, X 12M) subtracted from it? Please reconsider how this is formulated in the text. This doesn't work as written.*

That is correct. We have improved the clarity of the manuscript by writing 'X 0M' instead of 'X' to reflect the fact that this refers to the variable 'X' in the current month.
* * *
*-Table 2: To help choose which variables were useful for the ML algorithm (and which might only be contributing to overfitting), why did the authors not try applying one of the most standard techniques available such as recursive feature elimination with crossvalidation (available in scikit-learn, which the authors are already using (L132); Pedregosa et al. 2011)? These techniques could help get around arbitrary choices about the number of predictors, e.g. why 15 and not 10 or 20?*

Part of our original intention was to evaluate how different importance measures affect the choice of variables. While RFECV is of course possible to carry out in principle, this is usually easily done with the Gini importance which is quite cheap (computationally) to calculate, as it only considers data already seen during training. Unfortunately, this also means that RFECV fails to account for overfitting, as it only considers the training data when calculating feature importance (Meyer et al., 2019). In contrast, the different approaches we utilise to calculate feature importance (aiming to calculate a more robust metric) are computationally demanding, making an approach such as RFECV infeasible.

While we agree that the choice of 15 predictors is somewhat arbitrary, this is heuristically based on the slope of the feature importance plots (see Fig. 3). Looking at these feature importances, where the importance change is 'flat' (by inspection) after 15 variables, no additional information was being conveyed. Therefore, we decided to use this as our threshold.

Given this choice of 15 variables (and some constraints as discussed in the manuscript), we also carry out an iterative approach where we investigate different sets of variables to determine which combination of vegetation variables yields the best results. This was done to balance the computationally intensive nature of the CV calculations with the ability to answer the question "Which vegetation proxy is the most important?".
* * *
*- Why does Fig S2/2/3/7/etc. have straight line cutoffs? Top of Mexico, Australia, E of Iran. Plotting error? Real problem? I don't recall this being mentioned in the main text but it is in all figures. Fig 3-Also for v. low BA, what about grey instead of black? Would be easier to see the larger BA values...*

The 'block-shapes' are caused by our choice of AGB dataset, which we have mentioned in all relevant figure captions in the updated manuscript. We have also changed the way BA is plotted to more correctly highlight areas with missing data as shown in Fig. 4.

We have added grey shading to indicate regions with fire data availability, but where one or more of the other datasets is not available. Light grey indicates regions where mean BA is 0, with dark grey representing regions with nonzero mean BA.

Note that in addition to the shown changes in this figure, the colour scheme for plot c will also be revised to make the differences more apparent, and the number of divisions adjusted in order to make the plots clearer.
* * *
*- L 300- Are those the right refs? Both papers use Causality Analysis and not regression techniques as mentioned here. What aspects of those papers touches on inclusion of extra predictors and overfitting in a ML approach?*

The causality analyses in those papers are also based on regression techniques so that the same logic applies. In fact, causality methods are often used for predictions in

the same way as random forests (e.g. Kretschmer, Runge, and Coumou (2017)). The problem is always: there is a curse of dimensionality in high dimensional regression problems, and this can lead to overfitting if the dimensions are not controlled in some way. Our interpretation of random forests here is also causal by considering feature importances as a measure of driver importance behind the earth system process.

Both papers state, for example, that the explanative power of the machine learning approach they use (Causal Networks) decreases as additional, irrelevant variables are included. For example, Nowack et al. (2020) mentions "low detection power [. . .] if too many variables are used" and Runge et al. (2019) explains "Ideally, we want to condition only on the few relevant variables that actually explain a relationship".

We expect these observations to apply to our methodology, because inclusion of irrelevant variables may increase the likelihood of fitting to noise in the data, which, although likely somewhat mitigated by the bootstrapping employed by the random forest algorithm and cross-validation, may still present an issue.
* * *
*- p 12 has a lot of 'discussion' in the results section. Either have a 'results and discussion' section or keep them separate.*

As recommended by the reviewer, we have adapted the manuscript to have a combined 'Results and Discussion' section.
* * *
*- Fig 6 - can you not use sci notation for the numbers? It makes it easier to read. I am not sure if this warrants main text inclusion as it is very simple with almost no interesting structure. I would put this in supplement and move S2 or S3 into the main.*

These figures will be adapted to no longer make use of scientific notation in the tick labels.

While results shown in Fig. 6 (in the manuscript) may not be revolutionary or complex, they do illustrate one of the key findings – the ability to empirically determine and visualise intuitive interactions between variables. We therefore believe this figure still warrants inclusion in the main text.
* * *
*- L310 - what about masking for areas with longer fire return intervals to see if it [then] pops out as more important?*

Unfortunately, even considering the 20-year long MODIS record, we are strongly limited by the data available to us. Predictability in boreal ecosystems is expected to remain very limited because the return times are many times longer than the time series, so there is a very large stochastic component. Therefore, we don't currently expect this to be a viable analysis.
* * *
**References**

Abarca, Sergio F., Kristen L. Corbosiero, and Thomas J. Galarneau. 2010. 'An Evaluation of the Worldwide Lightning Location Network (WWLLN) Using the National Lightning Detection Network (NLDN) as Ground Truth'. Journal of Geophysical Research 115 (D18): D18206. https://doi.org/10.1029/2009JD013411.

Avitabile, Valerio, Martin Herold, Gerard B. M. Heuvelink, Simon L. Lewis, Oliver L. Phillips, Gregory P. Asner, John Armston, et al. 2016. 'An Integrated Pan-Tropical Biomass Map Using Multiple Reference Datasets'. Global Change Biology 22 (4): 1406–20. https://doi.org/10.1111/gcb.13139.

Bürgesser, Rodrigo E. 2017. 'Assessment of the World Wide Lightning Location Network (WWLLN) Detection Efficiency by Comparison to the Lightning Imaging Sensor (LIS): WWLLN Detection Efficiency Relative to LIS'. Quarterly Journal of the Royal Meteorological Society 143 (708): 2809–17. https://doi.org/10.1002/qj.3129.

Harris, I., P.D. Jones, T.J. Osborn, and D.H. Lister. 2014. 'Updated High‐resolution Grids of Monthly Climatic Observations – the CRU TS3.10 Dataset'. International Journal of Climatology 34 (3): 623–42. https://doi.org/10.1002/joc.3711.

Jolly, W. Matt, Mark A. Cochrane, Patrick H. Freeborn, Zachary A. Holden, Timothy J. Brown, Grant J. Williamson, and David M. J. S. Bowman. 2015. 'Climate-Induced Variations in Global Wildfire Danger from 1979 to 2013'. Nature Communications 6 (1): 7537. https://doi.org/10.1038/ncomms8537.

Kaplan, Jed O., and Hong-Kiu Lau. 2019. 'The WGLC Global Gridded Monthly Lightning Stroke Density and Climatology'. PANGAEA. https://doi.org/10.1594/PANGAEA.904253.

Kretschmer, Marlene, Jakob Runge, and Dim Coumou. 2017. 'Early Prediction of Extreme Stratospheric Polar Vortex States Based on Causal Precursors'. Geophysical Research Letters 44 (16): 8592–8600. https://doi.org/10.1002/2017GL074696.

Kühn, Ingolf, and Carsten F. Dormann. 2012. 'Less than Eight (and a Half) Misconceptions of Spatial Analysis'. Journal of Biogeography 39 (5): 995–98. https://doi.org/10.1111/j.1365-2699.2012.02707.x.

Meyer, Hanna, Christoph Reudenbach, Stephan Wöllauer, and Thomas Nauss. 2019. 'Importance of Spatial Predictor Variable Selection in Machine Learning Applications – Moving from Data Reproduction to Spatial Prediction'. Ecological Modelling 411 (November): 108815. https://doi.org/10.1016/j.ecolmodel.2019.108815.
Nowack, Peer, Jakob Runge, Veronika Eyring, and Joanna D. Haigh. 2020. 'Causal Networks for Climate Model Evaluation and Constrained Projections'. Nature Communications 11 (1): 1415. https://doi.org/10.1038/s41467-020-15195-y.

Pedregosa, F., G. Varoquaux, A. Gramfort, V. Michel, B. Thirion, O. Grisel, M. Blondel, et al. 2011. 'Scikit-Learn: Machine Learning in Python'. Journal of Machine Learning Research 12: 2825–30.

Ploton, Pierre, Frédéric Mortier, Maxime Réjou-Méchain, Nicolas Barbier, Nicolas Picard, Vivien Rossi, Carsten Dormann, et al. 2020. 'Spatial Validation Reveals Poor Predictive Performance of Large-Scale Ecological Mapping Models'. Nature Communications 11 (1): 4540. https://doi.org/10.1038/s41467-020-18321-y.

Roberts, David R., Volker Bahn, Simone Ciuti, Mark S. Boyce, Jane Elith, Gurutzeta Guillera‐Arroita, Severin Hauenstein, et al. 2017. 'Cross-Validation Strategies for Data with Temporal, Spatial, Hierarchical, or Phylogenetic Structure'. Ecography 40 (8): 913–29. https://doi.org/10.1111/ecog.02881.

Rodger, C. J., J. B. Brundell, R. L. Dowden, and N. R. Thomson. 2004. 'Location Accuracy of Long Distance VLF Lightning Locationnetwork'. Annales Geophysicae 22 (3): 747–58. https://doi.org/10.5194/angeo-22-747-2004.

Runge, Jakob, Peer Nowack, Marlene Kretschmer, Seth Flaxman, and Dino Sejdinovic. 2019. 'Detecting and Quantifying Causal Associations in Large Nonlinear Time Series Datasets'. Science Advances 5 (11): eaau4996. https://doi.org/10.1126/sciadv.aau4996.

Thurner, Martin, Christian Beer, Maurizio Santoro, Nuno Carvalhais, Thomas Wutzler, Dmitry Schepaschenko, Anatoly Shvidenko, et al. 2014. 'Carbon Stock and Density of Northern Boreal and Temperate Forests'. Global Ecology and Biogeography 23 (3): 297–310. https://doi.org/10.1111/geb.12125.

[Figure]

**Fig. 1.** R2 scores for burnt area (BA) prediction on test samples for different exclusion radii around the test samples. Each line represents the R2 score for around 333 individual test samples.

[Figure]

**Fig. 2.** The proportion of filled samples for FAPAR, with yellow indicating that all occurrences of a given month at a given location were filled and purple indicating no filling was done.

[Figure]

**Fig. 3.** Sorted variable importance metrics (Gini, PFI, SHAP, and LOCO) for the ALL model, with the highest variable importance according to each metric at the top. Note alternative metrics are greyed out.

[Figure]

**Fig. 4.** (a) Average observed (Ob.) BA derived from the GFED4 BA dataset. (b) Out-of-sample predictions (Pr.) by the ALL model. (c) Relative prediction error of the ALL model.

---

## Author Comment (AC4) · 8 Apr 2021

Dear René Orth,

Thank you for your thorough review which will surely enhance the quality of this work.

Referee comments are cited in *italics* and author's responses in normal font. Responses are separated by horizontal lines.
* * *
*This study investigates the role of antecedent fuel and moisture conditions for global*

*temporal and spatial fire patterns represented by burnt area. Using a suite of random forest models with different sets of explanatory variables, the authors show that both antecedent moisture and fuel conditions are relevant for accurately modelling/predicting observed burnt areas. Thereby, the time scales extend over a few months prior to the fire, with pronounced variations across biomes.*

*Recommendation: I think the paper requires moderate revisions.*

*This is an interesting analysis that is both relevant to the readership of Biogeosciences and a timely contribution to the ecohydrology-fire community. Legacy effects undoubtedly affect wildfire dynamics and can be a potential source of difficulties of state-of-the-art models to accurately capture fire dynamics across time scales. The machine learning approach in concert with various ecological and meteorological datasets is therefore well suited to study the underlying relationships without prior assumptions to finally provide valuable insights for the development of physically-based models. However, I have some concerns regarding the robustness of the analysis with respect to the gap filling strategy, the employed fire dataset and the relatively short analysis time period, which should to be addressed before the paper is published in Biogeosciences.*
* * *
*General comments:*
* * *
*(1) While I recognize the necessity to perform gap filling for the random forest approach in this study, I do not really like the strategy. Persistent gaps are filled using minimum values which in the case of soil water index would produce artifical droughts. While I actually do not fully understand the difference between transient and persistent gaps I agree with the authors that applying a regression-based can be suitable to fill short gaps of a few months. Nevertheless, and especially for the longer gaps extending across several consecutive months I think at least the role of the gap filling for the final*

*conclusions needs to be tested. This could be done by additionally using an alternative gap filling strategy, or by adding random noise to the gap filled values which could be scaled by the typical inter-annual dynamics of the respective month-of-year or season of the concerned metric.*

*(2) It is known that there are differences between fire datasets. To illustrate the robustness of the findings of this study, it would be helpful to re-compute selected key figures with the MODIS-based ESA CCI fire dataset (Chuvieco et al. 2018).*

*(3) I agree with the authors that the relatively short analysis time period could have a impact on the results, particularly with respect to the long legacy effects. In this context, as lightning data which limits the available time period is not employed in all experiments with 15 predictors, they could be performed with more input data covering a longer time period.*

*(4) I really like that different metrics are jointly used to quantify the importance of the predictors in a robust way. It would be great if the authors could add some information in the (dis?)agreement of the results between the individual importance metrics, also to inform similar future analyses.*

*I do not wish to remain anonymous - Rene Orth.*

(1) The algorithm used to discern the location of 'permanent' as opposed to 'transient' gaps utilises the amount of missing information for a specific month at each location. For example, if a certain grid cell was missing data for more than 50% of all Decembers in the record, these gaps in December would be treated as a permanent gap and therefore subject to filling by minimum values. Remaining gaps are treated as transient and therefore filled using the regression model outlined in the text.

Use of a different gap filling mechanism (Fig. 1 on the left; temporal nearest-neighbour gap-filling) yields very similar results (this applies to ALE plots, too). This simple nearest-neighbour gap-filling approach works by considering timeseries at each location and filling gaps by using the temporally closest valid sample at that location. Of the two approaches, we have decided to use the season-trend model with minima filling because it represents a more physical solution, which is based on an approach previously used for vegetation variables (see Beck et al. 2006), for which one would expect minima to occur during winter.

Additionally, we have quantified how many samples were filled using minimum values outside of winter, and as can be seen in Fig. 2, virtually no samples are filled using minimum values outside of winter in the northern extreme latitudes.

Regarding the filling of SWI, this should not have a big influence on the final results because we are not using antecedent values of SWI in any case. Since we don't expect fires during the winter, having (by necessity of gap filling) unphysical values of SWI in the winter should not affect results where relevant for our analysis.

(2 & 3) In our revised manuscript, we present an analysis of monthly data (as opposed to climatological averages) for the time period 11-2000–12-2019 (230 months), which also expands upon the previously limited temporal range. A different burnt area dataset, MODIS MCD64CMQ BA, was also used for this run. This experiment, the 15VEG_FAPAR_MON experiment, otherwise uses the same variables as the 15VEG_FAPAR experiment.

These results show decreased R2 values, which is to be expected given the higher variance of this data:

- Using random cross-validation; Test R2: 0.45, OOB R2: 0.45

- Excluding the years 2009-2012 (including 2012); Test R2: 0.38, OOB R2: 0.45

- Excluding the years 2016-2019 (including 2019); Test R2: 0.41, OOB R2: 0.45

For this temporal cross-validation either period 2009-2012 or period 2016-2019 is left out for testing.

The main relationships identified by the model did not change appreciably, however, showing that there is no temporal change that is important. The spatial patterns are dominant, since the models behave very similarly when comparing climatological and monthly data, and the main commonality between those data is the geographical pattern.

For example, the FAPAR ALE plots for the 15VEG_FAPAR (climatological) and 15VEG_FAPAR_MON (monthly, expanded temporal range) models are shown in Figures 3 and 4, respectively.

These ALEs are highly consistent. Note that the smaller variability in the second plot (monthly analysis) arises from the fact that a constant proportion of bootstrap samples is used to construct the shaded region, and since the monthly data has more samples, these random samples are larger.

Other variables are similarly consistent, e.g. DD 3M as shown in Figures 5 and 6.

We expect results for the MODIS-based ESA CCI fire dataset to be consistent with the above, since it is based on the same underlying product as the MODIS MCD64CMQ dataset.

(4) The results shown in Fig. 1 are also indicative of the disagreement between the different importance metrics that were used. We will include an analogous figure (sorted by combined importance instead of Gini importance) in the updated supplementary materials, which will show the level of (dis-)agreement between the different metrics.
* * *
*Specific comments:*
* * *
*line 8: this should be "simulated burnt area" I guess*

Correct, this has now been rectified in the abstract.

*25/26: Here you could cite O et al. (2020) and/or other previous studies on related topics.*

This part of the introduction has been rewritten to include more references to previous studies of the effect of antecedent productivity, moisture, and fuel build-up on fire activity:

"A number of regional and global studies have indicated the importance of antecedent fuel build-up for BA. For example, links between fire activity and antecedent productivity have been found in South Africa (Van Wilgen et al., 2000), central Australia (Griffin et al., 1983), grass and shrublands in the western US (Littell et al., 2009; Westerling et al., 2003; Swetnam and Betancourt, 1998), NSW Australia (for bushfire fuel) (Jenkins et al., 2020), and southern Africa (Archibald et al., 2009). Global studies have identified similar relationships (a positive relationship between pre-season productivity and fire activity in the following dry season) in some dry areas. By studying the correlation between growing period (i.e. antecedent) soil moisture and fire activity, Krawchuk and Moritz (2011) found fire activity in dry regions to be related to antecedent productivity. Similarly, van der Werf et al. (2008) found a similar relationship for arid ecosystem (e.g. N. AUS), where antecedent wet conditions coupled with instantaneous drying were important. Other global studies have identified northern Australia as obeying this relationship too (Randerson et al., 2005; Spessa et al., 2005). In a more recent global analysis, O et al. (2020) found that for arid regions, wet anomalies (soil moisture) lead to increased fire later in the year by increasing fuel loads and biomass. Thus, it is clear that a better understanding of the timescales of fuel accumulation, the interaction between biophysical drivers and fuel build-up, and the effects of antecedent weather conditions on both fuel loads and fuel drying is needed in order to improve predictions of BA."

*line 66: What do you mean here with "visualization techniques"?*

Visualisation techniques here refers to the use of ALE plots and SHAP value plots in order to visualise the relationships between the predictor variables and BA and the interactions between the different predictor variables. This has been made clearer in the Introduction.
* * *
*line 117-125: So this means you are using anomalies in the case of antecedent values for DD and the vegetation productivity proxies but absolute values in the case of the current variables? While I can understand the motivation for the removal of the seasonal cycle, I feel this is inconsistent. Why not give it all the random forest model in the ALL analysis, i.e. absolute and anomaly versions of DD and the vegetation productivity proxies at current and antecedent times, and let the model decide which of these are most relevant? This would seem more objective to me.*

We are only using anomalies for very large antecedent shifts $\geq$12 months. The reasoning for this was heuristically based on our desire to keep results interpretable, as opposed to improving the model's performance. This is because we expect anomalies with respect to the current year's seasonal cycle to be more interpretable as opposed to the absolute values, which are bound to be quite similar to the current year's value due to their large autocorrelation (globally, on average). Since the model is able (in theory) to capture arbitrary, non-linear, relationships between predictor variables, the inclusion of absolute values or anomalies should not affect the performance of the model (or, consequently, the feature importances) if the model has access to both the values for $\geq$12 months and current year's values (i.e. <12 months). The model should be able to compute the anomalies itself if these were indeed significant predictors (which, as it turns out, for the most part they are not). The same also applies in reverse, i.e. the model should be able to reconstruct the absolute values from the anomalies since it has access to the current year's variables which were originally used to compute the

anomalies in the first place.
* * *
*line 139-141: Wouldn't it be more straightforward to use the OOB score for this, instead of dividing the dataset into training and validation parts while it is relatively short anyway?*

We recognise that looking just at the training (in-the-bag) and validation scores is not valid, which is why we are also including the OOB scores in the updated manuscript.

We are showing OOB results (see Fig. 7) because they are a good indication of how the model is expected to generalise to unseen data (Fox et al., 2017). However, a separate validation dataset is still needed in analyses like ours (see also e.g. Joshi and Sukumar (2021)) because we are optimising the model using the training data. Therefore, even the OOB data which the model 'technically' has never seen influence the choice of variables, for example. Thus, a truly unseen dataset, the validation dataset, is reserved for a more unbiased evaluation of how the model should generalise.
* * *
*line 142-143: Please add a comment why the random forest model is not re-calibrated for each experiment where different (numbers of) predictors are used.*

The random forest models are not individually calibrated because this would be too computationally demanding. A comment to this end has been added to the manuscript where we introduce the models and when talking about potential future work.
* * *
*lines 180-181: Why only the first 300'000?*

The SHAP interaction values were only calculated for a limited number of samples because of the very large computational effort required to do so. However, since a discussion of the SHAP interaction values is beyond the scope of the current work, we

will omit this statement from the manuscript for clarity.
* * *
*line 228: I agree with the approach to focus on the most relevant predictors, but why did you decide on using 15 rather even fewer which could probably reduce overfitting even more while still preserving most of the model skill?*

While we agree that the choice of 15 predictors is somewhat arbitrary, this is heuristically based on the slope of the feature importance plots (see Fig. 8). Looking at these feature importances, where the importance change is 'flat' (by inspection) after 15 variables, no additional information was being conveyed. Therefore, we decided to use this as our threshold.

Given this choice of 15 variables (and some constraints as discussed in the manuscript), we also carry out an iterative approach where we investigate different sets of variables to determine which combination of vegetation variables yields the best results. This was done to balance the computationally intensive nature of the CV calculations with the ability to answer the question "Which vegetation proxy is the most important?".
* * *
*line 252: I think the FAPAR impact is strongest at high levels rather than intermediate levels.*

That is correct. This sentence has been rephrased to clarify the original intention: the importance of FAPAR changes most rapidly at intermediate levels of FAPAR. This is important to clarify, since it establishes that the relationship of current FAPAR and BA is described by two plateaus at extreme values, combined with a rapid transition through intermediate values.
* * *
*lines 273-277: I think this is a particularly nice result which could be more highlighted in the abstract or conclusions.*

The empirically discovered interactions have now been mentioned both in the abstract and conclusions.
* * *
*line 324-326: Couldn't it be a solution to test the inclusion of antecedent BA as a predictor in the random forest model?*

We expect antecedent burning in previous months to be reflected in the vegetation variables used in our studies, e.g. a decrease in FAPAR following a fire. The problem we discuss in lines 324–326, however, would not be addressed by knowledge of the burnt area in previous months, because here we are concerned with burnt area within the current month and its effects on our analysis. As we discuss in the manuscript, the monthly timescales we are working on prevent us from clearly resolving the effects of fires in the current month.
* * *
*Figures 2 and 3: More colors are needed for the color bars to enable a finer distinction of the spatial patterns.*

More colours will be added to the colour bar.
* * *
*Figures 4 and 5: It could be informative to add uncertainty ranges to the curves, for example by re-running the random forest models many times.*

We originally felt that inclusion of such plots might introduce too many visual elements. However, we have now added uncertainty ranges to Fig. 4 (in the manuscript), generated by repeated bootstrapping of the training data before generating the plots. This is shown in Fig. 9 (in the current document).

However, we believe that due to the large number of curves present in Fig. 5 (in the manuscript), adding yet more elements to this plot is infeasible without an undue enlargement of the plot.
* * *
*Figure 5, caption: "LAI" should probably be removed in "First-order LAI ALEs"; furthermore it is not explained what is meant by first-order and second-order.*

That is correct and has now been rectified. The distinction between first-order and second-order ALE plots had previously been defined in the Methods section, albeit using the terms 1D and 2D instead of first-order and second-order, respectively. We have now removed this inconsistency to make this clearer.
* * *
*Figures 2-6: Please adapt the value labels of the axes and color bars to avoid the use of the exponent term to improve readability.*

We will remove the exponent terms in the labels (see Fig. 9).
* * *
*Figure 6: Why are there darker colors surrounding the gray area in the upperleft corner?*

These darker colours are the result of the overlap between the grey areas representing missing data and the colours representing the ALE plot itself. This overlap results from the fact that ALEs are calculated for each bin edge (thus being plotted as $\frac{1}{2}$ in one cell and $\frac{1}{2}$ in the next), while the missing data applies to entire bins (i.e. between the two bin edges, which is what the tick marks on the plot correspond to). Therefore, the rectangles indicating missing data end up being shifted by $\frac{1}{2}$ of the bin size relative to the ALE data. This partial overlap is also meant to indicate that the affected ALE values are less reliable than others.

*Figure 7: Why is there no data in the southern half of Australia?*

This spatial artifact is a result of our chosen AGB dataset. This has now been stated for all figures exhibiting these artifacts.

**References**

Archibald, Sally, David P. Roy, Brian W. van Wilgen, and Robert J. Scholes. 2009. 'What Limits Fire? An Examination of Drivers of Burnt Area in Southern Africa'. Global Change Biology 15 (3): 613–30. https://doi.org/10.1111/j.1365-2486.2008.01754.x.

Beck, Pieter S. A., Clement Atzberger, Kjell Arild Høgda, Bernt Johansen, and Andrew K. Skidmore. 2006. 'Improved Monitoring of Vegetation Dynamics at Very High Latitudes: A New Method Using MODIS NDVI'. Remote Sensing of Environment 100 (3): 321–34. https://doi.org/10.1016/j.rse.2005.10.021.

Chuvieco, Emilio, Joshua Lizundia-Loiola, Maria Lucrecia Pettinari, Ruben Ramo, Marc Padilla, Kevin Tansey, Florent Mouillot, et al. 2018. 'Generation and Analysis of a New Global Burned Area Product Based on MODIS 250 m Reflectance Bands and Thermal Anomalies'. Earth System Science Data 10 (4): 2015–31. https://doi.org/10.5194/essd-10-2015-2018.

Fox, Eric W., Ryan A. Hill, Scott G. Leibowitz, Anthony R. Olsen, Darren J. Thornbrugh, and Marc H. Weber. 2017. 'Assessing the Accuracy and Stability of Variable Selection Methods for Random Forest Modeling in Ecology'. Environmental Monitoring and Assessment 189 (7): 316. https://doi.org/10.1007/s10661-017-6025-0.

Griffin, GF, NF Price, and HF Portlock. 1983. 'Wildfires in the Central Australian Rangelands, 1970-1980.' Journal of Environmental Management 17 (4): 311–23.

Jenkins, Meaghan E., Michael Bedward, Owen Price, and Ross A. Bradstock. 2020. 'Modelling Bushfire Fuel Hazard Using Biophysical Parameters'. Forests 11 (9): 925. https://doi.org/10.3390/f11090925.

Joshi, Jaideep, and Raman Sukumar. 2021. 'Improving Prediction and Assessment of Global Fires Using Multilayer Neural Networks'. Scientific Reports 11 (1): 3295. https://doi.org/10.1038/s41598-021-81233-4.

Krawchuk, Meg A., and Max A. Moritz. 2011. 'Constraints on Global Fire Activity Vary across a Resource Gradient'. Ecology 92 (1): 121–32. https://doi.org/10.1890/09-1843.1.

Littell, Jeremy S., Donald McKenzie, David L. Peterson, and Anthony L. Westerling. 2009. 'Climate and Wildfire Area Burned in Western U.S. Ecoprovinces, 1916–2003'. Ecological Applications 19 (4): 1003–21. https://doi.org/10.1890/07-1183.1.

O, Sungmin, Xinyuan Hou, and Rene Orth. 2020. 'Observational Evidence of Wildfire-Promoting Soil Moisture Anomalies'. Scientific Reports 10 (1): 11008. https://doi.org/10.1038/s41598-020-67530-4.

Randerson, J. T., G. R. van der Werf, G. J. Collatz, L. Giglio, C. J. Still, P. Kasibhatla, J. B. Miller, J. W. C. White, R. S. DeFries, and E. S. Kasischke. 2005. 'Fire Emissions from C3 and C4 Vegetation and Their Influence on Interannual Variability of Atmospheric CO2 and $\Delta$13CO2'. Global Biogeochemical Cycles 19 (2). https://doi.org/10.1029/2004GB002366.

Spessa, Allan, Bevan McBeth, and Colin Prentice. 2005. 'Relationships among Fire Frequency, Rainfall and Vegetation Patterns in the Wet–Dry Tropics of Northern Australia: An Analysis Based on NOAA-AVHRR Data'. Global Ecology and Biogeography 14 (5): 439–54. https://doi.org/10.1111/j.1466-822x.2005.00174.x.

Swetnam, Thomas W., and Julio L. Betancourt. 1998. 'Mesoscale Disturbance and Ecological Response to Decadal Climatic Variability in the American Southwest'. Journal of Climate 11 (12): 3128–47. https://doi.org/10.1175/1520-0442(1998)011<3128:MDAERT>2.0.CO;2.

Van Wilgen, B. W., H. Biggs, S. P. O'Regan, and N. Mare. 2000. 'Fire History of the Savanna Ecosystems in the Kruger National Park, South Africa, between 1941 and 1996'. South African Journal of Science 96 (April). https://researchspace.csir.co.za/dspace/handle/10204/1890.

Werf, Guido R. van der, James T. Randerson, Louis Giglio, Nadine Gobron, and A. J. Dolman. 2008. 'Climate Controls on the Variability of Fires in the Tropics and Subtropics'. Global Biogeochemical Cycles 22 (3). https://doi.org/10.1029/2007GB003122.

Westerling, A. L., A. Gershunov, T. J. Brown, D. R. Cayan, and M. D. Dettinger. 2003. 'Climate and Wildfire in the Western United States'. Bulletin of the American Meteorological Society 84 (5): 595–604. https://doi.org/10.1175/BAMS-84-5-595.
* * *
[Figure]

**Fig. 1.** Comparison of feature importances using four metrics (Gini, PFI, SHAP, and LOCO) for the ALL model using nearest-neighbour filling on the left, and season-trend model filling on the right.

[Figure]

**Fig. 2.** The proportion of filled samples for FAPAR, with yellow indicating that all occurrences of a given month at a given location were filled and purple indicating no filling was done.

[Figure]

**Fig. 3.** First-order ALE plot showing the effect of instantaneous FAPAR on burnt area (BA) in the 15VEG_FAPAR model after accounting for all other variables.

[Figure]

**Fig. 4.** First-order ALE plot showing the effect of instantaneous FAPAR on burnt area (BA) in the 15VEG_FAPAR_MON model after accounting for all other variables.

[Figure]

**Fig. 5.** First-order ALE plot showing the effect of the 3-month antecedent dry-day period (DD 3M) on burnt area (BA) in the 15VEG_FAPAR model after accounting for all other variables.

[Figure]

**Fig. 6.** First-order ALE plot showing the effect of the 3-month antecedent dry-day period (DD 3M) on burnt area (BA) in the 15VEG_FAPAR_MON model after accounting for all other variables.

[Figure]

**Fig. 7.** Global climatological R2 scores for the different experiments. Note that although train R2 scores are shown here, train OOB scores are more indicative of model performance on unseen data.

[Figure]

**Fig. 8.** Sorted variable importance metrics (Gini, PFI, SHAP, and LOCO) for the ALL model, with the highest variable importance according to each metric at the top. Note alternative metrics are greyed out.

[Figure]

**Fig. 9.** First-order ALEs for different antecedent (< 1 yr) relationships with (a) FAPAR and (b) dry-day period (DD) in the 15VEG_FAPAR model, showing the underlying relationships with BA.

---

## Author Comment (AC5) · 8 Apr 2021

"We have also undertaken an analysis of monthly data (as opposed to climatological averages) for the time period 11-2000—12-2019 (230 months)" is meant to read "We have also undertaken an analysis of monthly data (as opposed to climatological averages) for the time period 11-2000–12-2019 (230 months)" instead.

Apologies for the confusion.

---

## Author Response (AR1)

**Response to reviews and proposed changes regarding bg-2020-409**

We thank all reviewers and the associate editor for their constructive feedback.

Previous responses to reviewers' comments can be found here: `https://bg.copernicus.org/preprints/bg-2020-409/`, but some comments will additionally be treated in more depth below.

As before, referee comments are cited in *italics* and author's responses in normal font. Responses are separated by horizontal lines. However, here we only discuss comments raised by the reviewers that warrant further discussion. All other comments will be treated as set out in our previous responses (AC1, AC2, AC3, and AC4, which can be found at the link above). Further, a list of major changes concludes this part of the document, followed by the tracked changes between the manuscript versions and the new supplementary materials for reference. Figure numbers and section references in our responses below refer to the updated documents which are included at the end.
* * *
**Additional responses to RC2**

*I am concerned about the cross-validation strategy employed here (L137. By randomly choosing the validation dataset the potential for spatial autocorrelation issues arises. This is well known in the literature (see Roberts et al. 2017; Ploton et al. 2020; Kühn and Dormann, 2012; Meyer et al. 2019). Here is a snippet from the abstract to the Roberts article: 'Ecological data often show temporal, spatial, hierarchical (random effects), or phylogenetic structure. Modern statistical approaches are increasingly accounting for such dependencies. However, when performing cross-validation, these structures are regularly ignored, resulting in serious underestimation of predictive error. One cause for the poor performance of uncorrected (random) cross-validation, noted often by modellers, are dependence structures in the data that persist as dependence structures in model residuals, violating the assumption of independence. Even more concerning, because often overlooked, is that structured data also provides ample opportunity for overfitting with non-causal predictors.'. Because the authors devote considerable space to discussion of these predictors, I think this issue is worth consideration. The authors also argue [that] the gap in R2 of the training-validation simulations gives an idea of the generalizability of the model - but that breaks down if there are spatial autocorrelation issues. Also there is some spatial structure in their biases (Fig S2) that could be coming from this issue. I would suggest adopting other CV strategies as outlined in the papers I list above.*

*As I am not yet convinced by their CV strategy, which is important as it impacts the results quite heavily, I suggest major revisions as I assume it will take a bit of work to demonstrate that the chosen CV strategy doesn't give misleading results.*

In addition to our previous response to RC2 and as already indicated therein, we have made it clear in our updated manuscript how we assert the robustness of our model. We have included the following in the updated methods section, which expands upon our previous response:

"However, this is only valid if there is no autocorrelation between the samples. We investigated the degree of spatial autocorrelation using a variogram of global GFED4 BA which informed a buffered leave-one-out (B-LOO) CV procedure following Ploton et al. [2020]. This was carried out to determine how much the autocorrelation that may be present influences the amount of potential overfitting. We did not employ the extrapolation-prevention procedure used in Ploton et al. [2020] because it led to the exclusion of significant areas like northern Australia and west Africa. The B-LOO CV was executed as follows, where $r_{max} = 50$ pixels and $N_t$ was chosen such that the number of potential training samples was guaranteed to be equal to or above $N_t$ for all $r \leq r_{max}$:

[revised manuscript text omitted]

represents a more physical solution; it is based on an approach previously used for vegetation variables [see Beck et al., 2006], for which one would expect minima to occur during winter. Indeed, as can be seen in Fig. S2, virtually no samples are being filled with minima outside of winter, and predominantly in the northern extreme latitudes. While our gap-filling methodology may yield unphysical values for non-vegetation variables like SWI, we do not expect the filling of SWI to have a big influence on the final results because we do not use antecedent values of SWI. Since we do not anticipate fires during the winter, having (by necessity of gap filling) potentially unphysical values of SWI in the winter should not affect results where relevant for our analysis."
* * *
**Additional response to RC3**

*(1) While I recognize the necessity to perform gap filling for the random forest approach in this study, I do not really like the strategy. Persistent gaps are filled using minimum values which in the case of soil water index would produce artificial droughts. While I actually do not fully understand the difference between transient and persistent gaps I agree with the authors that applying a regression-based can be suitable to fill short gaps of a few months. Nevertheless, and especially for the longer gaps extending across several consecutive months I think at least the role of the gap filling for the final conclusions needs to be tested. This could be done by additionally using an alternative gap filling strategy, or by adding random noise to the gap filled values which could be scaled by the typical inter-annual dynamics of the respective month-of-year or season of the concerned metric.*

We have updated our manuscript in order to further explain the gap-filling procedure and demonstrate its robustness:

"There are gaps in the SWI, FAPAR, LAI, SIF, and VOD datasets in winter months at latitudes above $\sim$60°N, and in the austral winter for southern South America, due to high solar zenith angles for FAPAR, LAI, and SIF and because of snow cover and frozen soil for SWI and VOD [see e.g. Moesinger et al., 2020]. There are also sporadic missing values in these datasets caused by e.g. cloud cover. Unfortunately, simple exclusion of the times lacking data is not possible for our analysis because we commonly rely on antecedent samples throughout. Thus, data gaps were filled using a two-step approach as in Forkel et al. [2017] in order to allow analysis of summer months at the affected locations. This approach differentiates between two gap types based on the amount of missing information for a specific month at each location.

First, 'persistent' gaps, defined as months for which $50\%$ or more of the observations across all years are missing, were filled using the minimum value observed at that location for the given predictor variable. We assume that this indicates missing data during the winter, since other causes for data gaps (e.g. cloud cover) are predominantly 'transient'. For example, if a certain grid cell was missing data for more than $50\%$ of all Decembers in the record, these gaps in December would be treated as persistent and therefore filled using minima.

Second, the remaining transient gaps were filled using season-trend regression models with four harmonic terms ($k = 4$) and without breakpoints. These models were fitted using ordinary least squares regression to the entire timeseries obtained during the first step, as mentioned before using data from January 2008 to April 2015 (or November 2000 to December 2019 for the monthly analysis). Cloud cover, which also affects detection in tropical and subtropical regions, is usually transient, and therefore filled using the regression models. Locations where no observations were available for $> 52$ months out of the total 88 months (regardless of whether such data gaps always occur in the same month, as for persistent gaps, or at any point throughout the year) were discarded in a trade-off between data quality and geographic extent. For the monthly analysis, locations were discarded given $> 138$ unavailable months (out of 239 months).

Use of a different gap filling mechanism (Fig. S1b; temporal nearest-neighbour gap-filling) yielded very similar results. This simple nearest-neighbour gap-filling approach used for the eventual ALL_NN model processes timeseries at a given location, filling gaps by using the temporally closest available samples at that location. Of the two approaches, we decided to use the season-trend model with minima filling because it represents a more physical solution; it is based on an approach previously used for vegetation variables [see Beck et al., 2006], for which one would expect minima to occur during winter. Indeed, as can be seen in Fig. S2, virtually no samples are being filled with minima outside of winter, and predominantly in the northern extreme latitudes. While our gap-filling methodology may yield unphysical values for non-vegetation variables like SWI, we do not expect the filling of SWI to have a big influence on the final results because we do not use antecedent values of SWI. Since we do not anticipate fires during the winter, having (by necessity of gap filling)

potentially unphysical values of SWI in the winter should not affect results where relevant for our analysis."

Also, we wish to clarify that transient gaps (as defined above, e.g. due to cloud cover) will always be filled using the season-trend regression model. This has been made clearer as in the response above.
* * *
**Changes to the manuscript**

Line numbers and other references in brackets below (e.g. L100) refer to the tracked changes document that is included further below.

1. Typographical errors and grammatical issues have been corrected.

2. (Fig. 6) Fig. 6 will remain in the manuscript.

3. Figure captions have been updated in order to pronounce key findings. The following sentences have been added to the captions of:

    (a) Fig. 1: "The CURR model is the only model that does not include antecedent conditions, and it performs much worse as a result. Despite the fact that the ALL model contains 50 predictors, while all other models contain just 15, it does not perform significantly better than the best models containing just 15 predictors (e.g. 15VEG_FAPAR). Note that although train $R^2$ scores are shown here, they are not indicative of model performance on unseen data, for which the shown train OOB scores should be used instead."

    (b) Fig. 2: "Despite the visual exaggeration of the errors, which are generally small, there is no overall pattern."

    (c) Fig. 4: "A clear difference between instantaneous and antecedent relationships can be seen in both cases, with instantaneous FAPAR limiting BA while antecedent FAPAR promotes BA, and vice versa for the dry-day period." Note that the enhancement of BA due to extreme droughts (extreme dry-day period) is apparent across time periods.

    (d) Fig. 5: "Notably, the relationship between LAI and BA is not modelled consistently by the CURR model (b), but relationships with BA are generally consistent across models otherwise."

    (e) Fig. 6: "It can be seen that the combined effect of FAPAR and FAPAR 1M on BA is positive if FAPAR is low while FAPAR 1M is high."

4. (L236) Regarding the choice of 15 predictor variables, we have updated the methods section:

    "The choice of 15 predictors was heuristically based on the slope of the feature importance plots (see Fig. S3), where, by inspection, the importance change is minimal after 15 variables. Thereafter, no additional information was being conveyed, so we decided to use this as our threshold. While use of the more rigorous recursive feature elimination with cross-validation (RFECV) would be possible in principle, this commonly makes use of the Gini importance owing to its ease of calculation, as it only considers data already seen during training. Unfortunately, this also means that RFECV fails to account for overfitting, as it only considers the training data when calculating feature importance [Meyer et al., 2019]. In contrast, the different approaches we jointly utilised to calculate a more robust feature importance metric are much more computationally demanding, making RFECV infeasible."

5. The title has been changed to "The Importance of Antecedent Vegetation and Drought Conditions as Global Drivers of Burnt Area".

6. The abstract has been updated to further pronounce the empirically discovered interactions and the fact that the current conditions relevant for the moisture-limited regions relate to fuel drying. It has also generally been rephrased.

7. Numerical output (and importance ranks) have been updated throughout the paper in response to renewed runs of the model resulting in different hyperparameters.

8. (L25) The introduction has been rephrased as detailed in AC2.

9. (L67) We have added the following paragraphs to the introduction which further detail previous relevant studies:

"A number of regional and global studies have indicated the importance of antecedent fuel build-up for BA. For example, links between fire activity and antecedent productivity have been found in South Africa [Van Wilgen et al., 2000], central Australia [Griffin et al., 1983], grass and shrublands in the western United States [Littell et al., 2009, Westerling et al., 2003, Swetnam and Betancourt, 1998], New South Wales, Australia, for bushfire fuel [Jenkins et al., 2020], and southern Africa [Archibald et al., 2009]. Global studies have identified similar relationships (a positive relationship between pre-season productivity and fire activity in the following dry season) in some dry areas. By studying the correlation between growing period (i.e. antecedent) soil moisture and fire activity, Krawchuk and Moritz [2011] found fire activity in dry regions to be related to antecedent productivity. Similarly, van der Werf et al. [2008] found a similar relationship for arid ecosystem (e.g. northern Australia), where antecedent wet conditions coupled with instantaneous drying were found to be important. Other global studies have also identified northern Australia as obeying this relationship [Randerson et al., 2005, Spessa et al., 2005]. In a more recent global analysis, O et al. [2020] found that for arid regions, wet anomalies (soil moisture) lead to increased fire later in the year by increasing fuel loads and biomass. Thus, it is clear that a better understanding of the timescales of fuel accumulation, the interaction between biophysical drivers and fuel build-up, and the effects of antecedent weather conditions on both fuel loads and fuel drying is needed in order to improve predictions of BA.

While other studies have used machine learning to explore fire drivers including the effect of antecedent productivity [e.g. Archibald et al., 2009, Forkel et al., 2017, Joshi and Sukumar, 2021], they have not explored the relationship between antecedent conditions (fuel load and drying) and fire in detail. Here we quantify the roles that antecedent vegetation productivity and aridity play relative to instantaneous conditions, the critical number of months that are most important for each, the shape of their relationships to BA, and the interactions between them. While the (relative) importance of antecedent variables has been investigated before [Bessie and Johnson, 1995], we aim to quantify this on a global scale. Since other climate factors, ignitions, and human activities also influence BA, we necessarily include these factors in our analysis. The use of a machine learning approach enables us to identify non-linear relationships and interactions between the drivers. This is then combined with analysis and visualisation techniques that provide insights into the modelled relationships while mitigating the effects of correlations among variables. Such insights include the effect of a particular driver on BA and the interactions between pairs of drivers."

10. (L90) The visualisation techniques used in our work have been described in more detail at the end of the introduction:

"The use of a machine learning approach enables us to identify non-linear relationships and interactions between the drivers. This is then combined with analysis and visualisation techniques that provide insights into the modelled relationships while mitigating the effects of correlations among variables. Such insights include the effect of a particular driver on BA and the interactions between pairs of drivers."

11. (L98) While introducing the datasets used, we now mention that "A longer time period from November 2000 to December 2019 was also considered in an analysis using fewer variables.", in order to introduce the new monthly analysis that was undertaken.

12. (L107) We have added an example dry-day calculation:

"A period contiguous with the previous month's dry-day period was concatenated such that the sum of both (number of days) was used to determine the longest period. For example, consider a 30-day long month with a 10-day long dry-day period at the beginning of the month, followed by a wetting precipitation event on day 11, and then a dry-day period for the following 19 days. This month has a dry-day period of 19 days. However, if the previous month were to terminate in a 10-day long dry-day period, these 10 days would be added to the initial 10-day dry-day period of the current month, thereby making this combined dry-day period the longest."

13. (L114) We have clarified that the WGLC data we are using (which is based on but not equivalent to WWLLN data) mainly detect cloud-to-ground strikes:

"We used the WGLC dataset [Kaplan and Lau, 2019] which provides counts of monthly lightning strikes. It is based on the World Wide Lightning Location Network (WWLLN) dataset, which mainly detects cloud-to-ground strikes [Rodger et al., 2004, Abarca et al., 2010], as opposed to LIS lightning data [Bürgesser, 2017]."

14. (L122) We have clarified how the AGB datasets we used were combined in regions where they overlap (which is by taking the mean):

"Tree AGB was obtained by mosaicking AGB datasets for the tropics [Avitabile et al., 2016, 1 km resolution] and northern forests [Thurner et al., 2014, 0.01° resolution] using the mean after resampling each to a common spatial resolution of 0.25°."

15. (L125) We mention our usage of an updated version of the HYDE 3.2 dataset. This is required for the new longer monthly analysis.

16. (Table 1) We have indicated datasets that are continually being updated in the updated Table 1.

17. (Table 1) We have added the MCD64CMQ BA dataset to the updated Table 1.

18. (Table 1) We have updated the table to reflect updated datasets.

19. (L185) We have used X 0M instead of X when discussing the calculation of antecedent anomalies to make it clearer that we are referring to the current variable.

20. (L199) We have updated our description of the hyperparameters used to reflect the updated state of our models.

21. (L204) We have mentioned that the number of split levels were limited both in order to limit overfitting and to enable SHAP value calculation.

22. (L259) We have included the following description of our monthly analysis in the methods section:

"In addition to the above climatological experiments, we also investigate monthly data for the time period 11-2000–12-2029 (230 months) using the 15VEG_FAPAR_MON model. To avoid the temporal limits of the GFED4 dataset (see Table 1) the MODIS MCD64CMQ [Giglio et al., 2018] BA dataset was used. Otherwise, this experiment uses the same variables as the 15VEG_FAPAR experiment with the exception of lightning, which was replaced with the similarly significant variable AGB (see Fig. S3) in order to enable processing of a longer time period. In addition to five-fold random CV as for the other models, the performance and generalisability of this model was also measured using temporal CV. Here, the model was trained on all samples excluding either all months within the years 2009–2012 (including 2012) or 2016–2019 (including 2019). Thereafter, the $R^2$ was measured on whichever years were excluded for training. Note that unless explicitly specified, all following methodological descriptions will relate to the climatological experiments as opposed to the monthly 15VEG_FAPAR_MON experiment."

23. (Table 2) We have added the 15VEG_FAPAR_MON and ALL_NN experiments to Table 2.

24. (Sect. 2.4) We have rephrased Sect. 2.4 in order to explain our calculations more clearly.

25. (L314) Instead of introducing 1D and 2D ALEs, we now uniformly refer to these as first-order and second-order ALEs.

26. (L320 onwards) We have combined the results and discussion sections.

27. (Fig. 2) We have updated all maps to include grey shading that indicates regions where BA data is available, but other datasets are not.

28. (L336) We have added an explanation regarding the absence of 0 predictions by the model:

"The model may struggle to predict 0 BA because the random forest model consists of many smaller decision trees. All 500 individual models would have to predict 0 to yield this value overall, which does not appear to occur given the stochastic nature of the training process. Failure to capture extreme events well is likely due to their rarity, resulting in the absence of comparable training data [see also e.g. Joshi and Sukumar, 2021]."

29. (L340) As indicated further above, we have included an explanation of the B-LOO CV procedure and its results.

30. (L366) We have updated our discussion of why antecedent conditions $\geq 1$ yr. may not be important predictors.

31. (L381) Paragraphs from the previous discussion section have been moved into the combined results and discussion section.

32. (Figure 4) Figure 4 has been updated to include uncertainties using shaded regions. An inset axis has also been added to (b) to further highlight the relationships between DD and BA. The updated figure caption makes it clear that we used $\sim 10\%$ of training data 100 times to construct the uncertainty ranges.

33. (L450) We have added a reference to a recent relevant study ([Joshi and Sukumar, 2021]) in our discussion of poor predictability in boreal regions (amongst others).

34. (L455) Our description of the relationship between FAPAR and BA has been corrected to indicate that it changes most rapidly at intermediate levels of FAPAR.

35. (L457) This sentence has been moved up in order to make the paragraph more coherent.

36. (L474) The results and discussion sections have been merged.

37. (L492) Results and discussion of our new monthly analysis have been added, in order to illustrate the robustness of our modelling approach given a variety of CV scenarios:

"Relationships between predictors and BA were also stable when considering the 15VEG_FAPAR_MON model (Fig. S12) which not only uses monthly instead of climatological data, but also a different BA dataset. Using random CV, a test $R^2$ of 0.501 and an OOB train $R^2$ of 0.498 were measured. Excluding the years 2009–2012, a test $R^2$ of 0.403 and an OOB train $R^2$ of 0.507 were measured. Excluding the final years 2016–2019, a test $R^2$ of 0.435 and an OOB train $R^2$ of 0.505 were measured. While these $R^2$ scores are lower than those observed for the previously discussed climatological analyses, they demonstrate that the model is able to robustly predict BA under multiple CV scenarios. Lower $R^2$ scores are also expected given the higher variance of this data. Additionally, the relationships identified by the model are highly consistent with the previous climatological analyses, showing that there is no temporal change that is important. The spatial patterns are dominant as the models behave very similarly when fit on climatological and monthly data; and the main commonality between those data is the geographical pattern. Note also that while lightning is omitted from this experiment in contrast to the climatological 15VEG_FAPAR experiment, lightning is also not present in the TOP15 model which performs similarly to the ALL and BEST15 models. Furthermore, as shown in Fig. S3, the importance of lightning and its replacement, AGB, are very similar."

38. (L515) A reference to van der Werf et al. [2008] has been added to contextualise the empirically discovered interactions.

39. (L535) The conclusions section has been expanded to further highlight previous studies, make the novel contributions in our paper clearer, and introduce potential future work.

40. (L626) An additional acknowledgement has been added due to the contribution of an updated HYDE dataset by Kees Klein Goldewijk.

[revised manuscript text omitted]

 These were then used to calculate the maximally significant timescales for the basis predictor variable  X (e.g. FAPAR)  using an average over antecedent months, a, weighted by maximum SHAP value magnitudes:

$$t_{\max,\text{X},\ell} = \frac{\sum_{\text{a},x} \text{a}|\text{SHAP}_{x,\ell}|}{\sum_{\text{
[revised manuscript text omitted]

**Table S1.** Variables used in the experiments. 'C' denotes current-month variables, 'all A' represents all antecedent months (1M–24M), and 1M represents one-month antecedent variables, with similar notation for other antecedent months.

| | DD | SWI | MaxT | DTR | Light-ning | CROP | POPD | HERB | SHRUB | TREE | AGB | VOD | FAPAR | LAI | SIF |
|---|---|---|---|---|---|---|---|---|---|---|---|---|---|---|---|
| ALL | C & all A | C | C | C | C | C | C | C | C | C | C | C & all A | C & all A | C & all A | C & all A |
| ALL_NN | C & all A | C | C | C | C | C | C | C | C | C | C | C & all A | C & all A | C & all A | C & all A |
| CURR | C | C | C | C | C | C | C | C | C | C | C | C | C | C | C |
| BEST15 | C, 1M, 3M, 6M, 9M | | C | C | C | C | C | | | | | | C, 1M | 3M | 6M, 9M |
| TOP15 | C, 1M, 3M, 9M | | C | | | C | C | | | | | C, 1M, 3M | C, 1M | 1M, 3M | C |
| 15VEG_FAPAR | C, 1M, 3M, 6M, 9M | | C | C | C | C | C | | | | | | C, 1M, 3M, 6M, 9M | | |
| 15VEG_FAPAR-_MON | C, 1M, 3M, 6M, 9M | | C | C | | C | C | | | | C | | C, 1M, 3M, 6M, 9M | | |
| 15VEG_LAI | C, 1M, 3M, 6M, 9M | | C | C | C | C | C | | | | | | | C, 1M, 3M, 6M, 9M | |
| 15VEG_SIF | C, 1M, 3M, 6M, 9M | | C | C | C | C | C | | | | | | | | C, 1M, 3M, 6M, 9M |
| 15VEG_VOD | C, 1M, 3M, 6M, 9M | | C | C | C | C | C | | | | | C, 1M, 3M, 6M, 9M | | | |
| CURRDD_FAPAR | C | C | C | C | C | C | | C | C | C | C | | C, 1M, 3M, 6M, 9M | | |
| CURRDD_LAI | C | C | C | C | C | C | | C | C | C | C | | | C, 1M, 3M, 6M, 9M | |
| CURRDD_SIF | C | C | C | C | C | C | | C | C | C | C | | | | C, 1M, 3M, 6M, 9M |
| CURRDD_VOD | C | C | C | C | C | C | | C | C | C | C | C, 1M, 3M, 6M, 9M | | | |

**Table S2.** Ranked importance of variables in the RF experiments according to the composite importance measure introduced in Sect. 2.4.

| | ALL | ALL_NN | TOP15 | CURR | 15VEG_FAPAR | 15VEG_LAI | 15VEG_SIF | 15VEG_VOD | CURRDD_FAPAR | CURRDD_LAI | CURRDD_SIF | CURRDD_VOD | BEST15 |
|---|---|---|---|---|---|---|---|---|---|---|---|---|---|
| 1 | DD | DD | FAPAR | DD | FAPAR | LAI | SIF | VOD 1M | FAPAR 1M | LAI | SIF | VOD 1M | FAPAR |
| 2 | FAPAR | FAPAR | MaxT | MaxT | DD | DD | DD | VOD | DD | LAI 1M | DD | VOD | DD |
| 3 | VOD 1M | VOD 3M | MaxT | TREE | FAPAR 1M | LAI 1M | MaxT | VOD | DD | DD | MaxT | VOD 3M | FAPAR 1M |
| 4 | VOD 3M | VOD 1M | SIF | VOD 3M | MaxT | LAI 3M | CROP | VOD 3M | FAPAR 6M | LAI 3M | SIF 3M | DD | LAI 3M |
| 5 | MaxT | MaxT | SIF | SWI | CROP | MaxT | SIF 6M | MaxT | FAPAR 6M | MaxT | SIF 6M | MaxT | CROP |
| 6 | SIF | SIF | DD 9M | DD 9M | CROP | CROP | DD 1M | VOD 9M | CROP | LAI 6M | SIF 9M | VOD 9M | MaxT |
| 7 | FAPAR 1M | VOD | LAI 1M | SIF | DD 3M | LAI 6M | DD 3M | DD 9M | FAPAR 3M | HERB | SIF 9M | VOD 6M | SIF 9M |
| 8 | CROP | FAPAR 1M | VOD | FAPAR | FAPAR 6M | DD 1M | SIF 9M | CROP | FAPAR 9M | LAI 9M | TREE | AGB | DD 1M |
| 9 | VOD | LAI 1M | DD 1M | HERB | FAPAR 9M | DD 3M | SIF 3M | DD 3M | HERB | CROP | CROP | DTR | POPD |
| 10 | DD 1M | DD 1M | FAPAR 1M | DTR | DD 9M | DD 9M | SIF 1M | VOD 6M | Lightning | Lightning | Lightning | Lightning | DD 9M |
| 11 | LAI 1M | LAI 3M | CROP | Lightning | DD 6M | LAI 9M | DD 6M | DD 1M | DTR | DTR | DTR | HERB | DD 6M |
| 12 | LAI 3M | CROP | LAI 3M | AGB | DD 9M | DD 6M | DD 9M | DD 6M | SWI | SHRUB | SWI | SHRUB | Lightning |
| 13 | DD 3M | POPD | VOD 1M | CROP | POPD | POPD | DTR | POPD | SHRUB | SWI | SHRUB | SWI | SIF 6M |
| 14 | DD 9M | DD 9M | DD 3M | SHRUB | Lightning | Lightning | Lightning | DTR | TREE | TREE | HERB | TREE | DD 3M |
| 15 | POPD | DD 3M | POPD | POPD | DTR | DTR | POPD | Lightning | AGB | AGB | AGB | CROP | DTR |
| 16 | SIF 9M | FAPAR 6M | | | | | | | | | | | |
| 17 | FAPAR 6M | SIF 9M | | | | | | | | | | | |
| 18 | LAI | DD 6M | | | | | | | | | | | |
| 19 | DD 6M | VOD 9M | | | | | | | | | | | |
| 20 | VOD 9M | Lightning | | | | | | | | | | | |
| 21 | Lightning | LAI | | | | | | | | | | | |
| 22 | DTR | FAPAR 9M | | | | | | | | | | | |
| 23 | AGB | FAPAR 3M | | | | | | | | | | | |
| 24 | SHRUB | DD △12M | | | | | | | | | | | |
| 25 | FAPAR 9M | DTR | | | | | | | | | | | |
| 26 | SIF 6M | AGB | | | | | | | | | | | |
| 27 | LAI 6M | LAI 6M | | | | | | | | | | | |
| 28 | SWI | SWI | | | | | | | | | | | |
| 29 | FAPAR 3M | SHRUB | | | | | | | | | | | |
| 30 | VOD △12M | SIF 6M | | | | | | | | | | | |
| 31 | DD △12M | LAI 9M | | | | | | | | | | | |
| 32 | SIF 3M | SIF 3M | | | | | | | | | | | |
| 33 | LAI 9M | VOD 6M | | | | | | | | | | | |
| 34 | VOD 6M | VOD △12M | | | | | | | | | | | |
| 35 | VOD △24M | SIF 1M | | | | | | | | | | | |
| 36 | VOD △18M | HERB | | | | | | | | | | | |
| 37 | SIF 1M | TREE | | | | | | | | | | | |
| 38 | LAI △24M | VOD △24M | | | | | | | | | | | |
| 39 | TREE | LAI △12M | | | | | | | | | | | |
| 40 | FAPAR △24M | SIF △24M | | | | | | | | | | | |
| 41 | HERB | DD △24M | | | | | | | | | | | |
| 42 | SIF △24M | LAI △24M | | | | | | | | | | | |
| 43 | FAPAR △12M | FAPAR △24M | | | | | | | | | | | |
| 44 | SIF △18M | SIF △18M | | | | | | | | | | | |
| 45 | SIF △12M | SIF △12M | | | | | | | | | | | |
| 46 | DD △24M | FAPAR △12M | | | | | | | | | | | |
| 47 | DD △18M | VOD △18M | | | | | | | | | | | |
| 48 | LAI △12M | LAI △18M | | | | | | | | | | | |
| 49 | FAPAR △18M | DD △18M | | | | | | | | | | | |
| 50 | LAI △18M | FAPAR △18M | | | | | | | | | | | |

[Figure]

**Figure S1.** Transformed variable importance metrics (Gini, PFI, SHAP, and LOCO) for the (a) ALL, (b) ALL_NN, and (c) 15VEG_FAPAR models. The 15 most important variables (with others omitted for clarity) are sorted by their combined importance with the most important on the left. Uncertainties using the standard deviation are indicated using shaded regions. The uncertainty magnitudes differ between the metrics due to the way they are calculated; SHAP values are calculated for every sample, Gini importances are calculated based on splits for individual decision trees, PFI calculations are repeated after permuting the original dataset, and LOCO importances are only calculated once. Therefore, based on the number of samples used for their calculation, the SHAP importances are expected to have the highest variance, followed by the Gini and then PFI importances and lastly the LOCO importances without any quantification of the error.

[Figure]

**Figure S2.** The fraction of filled samples for FAPAR (January 2008 to April 2015) at a given location for each month, with yellow indicating that all occurrences of a given month at a given location were filled and purple indicating no filling was done. Filling is mostly carried out in winter in northern latitudes.

[Figure]

**Figure S3.** Sorted variable importance metrics (Gini, SHAP, PFI, and LOCO) for the ALL model, with the highest variable importance according to each metric on the left. The dotted red line indicates the 15th variable. Uncertainties are calculated using the standard deviation and indicated using the shaded regions.

[Figure]

**Figure S4.** Out-of-sample BA predictions by the ALL model and corresponding observations. Note that logarithmic scales are used throughout except for the lower end of the x-axis, where a linear scale is used.

[Figure]

**Figure S5.** Mean difference between the out-of-sample observed (Ob.) and predicted (Pr.; by the ALL model) BA. The major spatial patterns follow the magnitude of mean BA (see Fig. 2a), with (on average) underprediction most prevalent in regions with large mean BA and vice versa. Note that sharp data availability boundaries (e.g. in western Asia, southern Australia) are introduced by the AGB dataset. Grey shading indicates regions with fire data availability, but where one or more of the other datasets is not available. Light grey indicates regions where mean BA is 0, with dark grey representing regions with non-zero mean BA.

[Figure]

**Figure S6.** PFI importances for the 15VEG_FAPAR model computed separately on the training and validation sets. The error bars originate from repeated shuffling of the investigated variable.

[Figure]

**Figure S7.** Variogram of mean GFED4 BA (June 1995 to December 2016) using all 237373 available samples. Semivariance can be seen to increase until ∼1000 km. Note that a logarithmic scale is used for the sample counts at the top.

[Figure]

**Figure S8.** $R^2$ scores for burnt area (BA) prediction on test samples for different exclusion radii around the test samples using the 15VEG_FAPAR model. Each of the 10 lines represents the $R^2$ score for 400 test samples computed for the shown radii, where each individual test sample is chosen randomly and surrounded by a circular region of ignored data that is not used for training, with varying radii as shown. The disagreement between the lines is indicative of the statistical uncertainty, regional variability, and potentially different degrees of model extrapolation.

[Figure]

**Figure S9.** Pearson correlations between all variables used in the analysis for the time period from January 2010 to April 2015. Especially large positive correlations exist between variable pairs separated by multiples of 12 months and between FAPAR, LAI, SIF, and VOD. The largest negative correlations are found between SWI and DD (instantaneous, 12 month, 24 month), SWI and DTR, and CROP and TREE.

[Figure]

**Figure S10.** Spatial distribution of individual peak combinations for SHAP values as in Fig. 7. The sign of the maximum effect on BA at a certain antecedent month is indicated in parentheses after each month. The peak combinations are shown here such that their ordering has no significance (i.e. $0(+)|3(+)$ equals $3(+)|0(+)$). Dominant antecedent periods are apparent from Fig. 7. Most clearly, the general limitation of BA by instantaneous DD in tropical and boreal regions is seen in (c), combined with the positive effect of instantaneous DD on burning in the remaining regions. The limitation of BA by instantaneous DD shown in (c) generally agrees with the enhancement of BA by three-month antecedent FAPAR shown in (b), as well as the enhancement by one-month antecedent DD in (d). Note that sharp data availability boundaries (e.g. in western Asia, southern Australia) are introduced by the AGB dataset. Grey shading indicates regions with fire data availability, but where one or more of the other datasets is not available. Light grey indicates regions where mean BA is 0, with dark grey representing regions with non-zero mean BA.

[Figure]

**Figure S11.** First-order LAI ALEs for different lags ($< 1$ yr) from all relevant modelling experiments for the relationships between BA and SIF (left hand columns) and VOD (right hand columns). Evenly spaced quantiles were used in the construction of the plots.

[Figure]

**Figure S12.** First-order ALE plots showing the effect of FAPAR (a, b) and the 3-month antecedent dry-day period (DD 3M; c, d) on burnt area (BA) in the 15VEG_FAPAR model (a, c) and the 15VEG_FAPAR_MON model (b, d) after accounting for all other variables. The shaded regions represent the standard deviation around the mean of 100 ALEs each using 122567 random samples of the training data.

[Figure]

**Figure S13.** Second-order ALE plot showing the combined zeroth order (mean), first order, and second order modelled effects of FAPAR and FAPAR 1M on BA from the 15VEG_FAPAR model, taking into account all other variables. Grey boxes indicate missing data. The diagonal structure of the sample count matrix demonstrates the correlation between these variables. Evenly spaced quantiles were used in the construction and labelling of the plots.

[Figure]

**Figure S14.** Second-order ALE plot showing the combined zeroth order (mean), first order, and second order modelled effects of DD and FAPAR on BA from the 15VEG_FAPAR model, taking into account all other variables. The diagonal structure of the sample count matrix demonstrates the anticorrelation between these variables. Evenly spaced quantiles were used in the construction and labelling of the plots.